# OBI-BENCH: CAN LMMS AID IN STUDY OF ANCIENT SCRIPT ON ORACLE BONES?

**Zijian Chen**[1], **Tingzhu Chen**[2*], **Wenjun Zhang**[1], **Guangtao Zhai**[1*]

[1]Institute of Image Communication and Information Processing, Shanghai Jiao Tong University
[2]School of Humanities, Shanghai Jiao Tong University

```
{zijian.chen, tingzhuchen, zhangwenjun, zhaiguangtao}@sjtu.edu.cn
```
*Corresponding authors
https://github.com/zijianchen98/OBI-Bench

## ABSTRACT

We introduce **OBI-Bench**, a holistic benchmark crafted to systematically evaluate large multi-modal models (LMMs) on whole-process oracle bone inscriptions (OBI) processing tasks demanding expert-level domain knowledge and deliberate cognition. OBI-Bench includes 5,523 meticulously collected diverse-sourced images, covering five key domain problems: recognition, rejoining, classification, retrieval, and deciphering. These images span centuries of archaeological findings and years of research by front-line scholars, comprising multi-stage font appearances from excavation to synthesis, such as original oracle bone, inked rubbings, oracle bone fragments, cropped single characters, and handprinted characters. Unlike existing benchmarks, OBI-Bench focuses on advanced visual perception and reasoning with OBI-specific knowledge, challenging LMMs to perform tasks akin to those faced by experts. The evaluation of 6 proprietary LMMs as well as 17 open-source LMMs highlights the substantial challenges and demands posed by OBI-Bench. Even the latest versions of GPT-4o, Gemini 1.5 Pro, and Qwen-VL-Max are still far from public-level humans in some fine-grained perception tasks. However, they perform at a level comparable to untrained humans in deciphering tasks, indicating remarkable capabilities in offering new interpretative perspectives and generating creative guesses. We hope OBI-Bench can facilitate the community to develop domain-specific multi-modal foundation models towards ancient language research and delve deeper to discover and enhance these untapped potentials of LMMs.

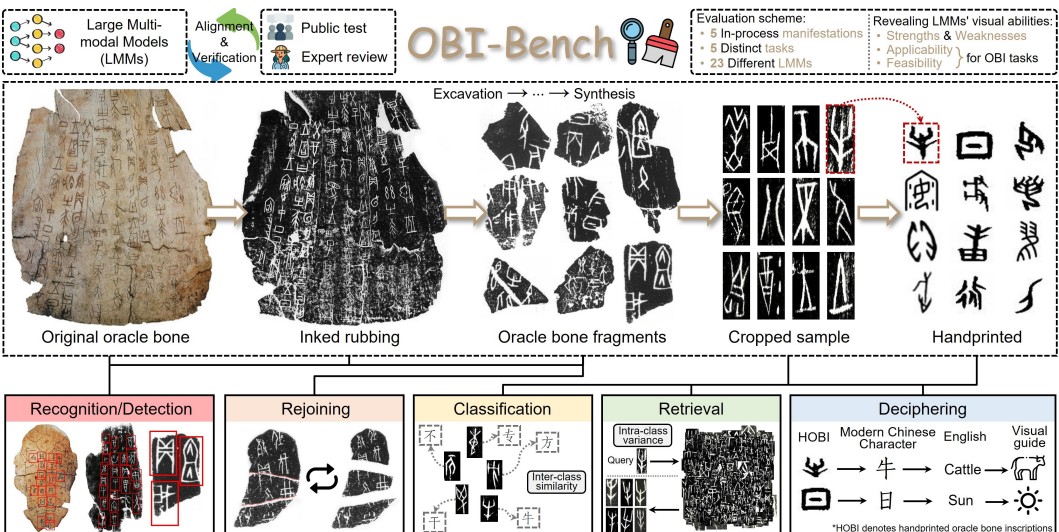

Figure 1: Overview of the **OBI-Bench**. OBI-Bench presents five in-process tasks: 1) recognition: locating dense oracle bone characters from original oracle bone or rubbings; 2) rejoining: reconstructing fragmented text fragments into coherent texts; 3) classification: categorizing individual characters into their respective meanings; 4) retrieval: returning relevant results according to the given query OBI images; 5) deciphering: interpreting the OBI for historical and cultural investigation.

# 1 INTRODUCTION

Oracle bone inscriptions (OBI) have been recognized as valuable records of making divination and praying to gods by the late Shang people from 1400 B.C. to 1100 B.C., in addition to being the earliest evidence of a Chinese writing system (UNESCO, 2017). Their discovery and interpretation provide a unique window for evidence of the thought, language, society, and history of past civilizations. All along, there have been several challenges from the excavation to the interpretation of OBI. First, after thousands of years buried in the ground, the original OBIs are inherently eroded, fragmented, and susceptible to damage. Recognizing oracle bone characters from scattered pieces not only helps in initially determining the existence of the characters but also aids in their digital archiving with minimal physical handling. Second, the rejoining work of OBI is an indispensable prerequisite for follow-up research, which restores the original appearance of oracle Bones and offers complete and accurate information for OBI scholars (Zhang et al., 2022). Current rejoining procedures are non-trivial and involve time-consuming and specialized workflows (Men, 2008), thus posing demands to develop automatic or AI-assisted rejoining techniques. Third, OBIs exhibit a wide range of stylistic variations in font and normally suffer from various types of noise in real rubbings (*e.g.*, *stroke broken*, *edge-cracked*, and *spindles*), making it arduous to distinguish from distortions and accurately classify them. Fourth, OBI retrieval, as a core component for building large-scale databases, its difficulty lies in differentiating the inter-class and intra-class discrepancies of oracle bone characters. Fifth, among the thousands of OBIs that have been excavated, over half of them remain undeciphered (Wang & Deng, 2024). The traditional methods for oracle character deciphering typically involve a combination of historical linguistics and archaeological context. More recently, lots of efforts (Jiang et al., 2023; Gao et al., 2024; Qiao et al., 2024; Guan et al., 2024b) have been made to leverage the representation capability of neural networks so as to establish correspondences between OBIs and modern Chinese. However, many characters have evolved in meaning through cultural and historical changes, and not all oracle bone characters can be translated into modern Chinese with synonymous expressions. Another problem lies in the absence of a comprehensive corpus and the lack of OBI-encoding systems, which limits the applicability of language models. These five tasks are crucial steps towards placing an OBI both in history and within the world of the people who read and study it.

Nowadays, the emergent large multi-modal models (LMMs) have brought opportunities for solving multidisciplinary tasks with powerful visual perception, understanding, and reasoning abilities (Tian et al., 2024; Yue et al., 2024; Caffagni et al., 2024; OpenAI, 2024b). Meanwhile, the ability of LMMs remains unclear on fine-grained perception and cognition (*i.e.*, *high-level perception*), which play significant roles in interpreting ancient texts and their associated tasks on image material pre-processing. The perception at different granularities is strongly associated with a wide range of OBI tasks, such as distinguishing noise from real oracle bone character recognition (Wang et al., 2022), marginal sealability checking in rejoining (Zhang et al., 2021c) and component-level structural decomposition in retrieval (Hu et al., 2024). Henceforth, it is worth evaluating the current abilities of LMMs in fine-grained perception and cognition based on actual needs to relieve extensive human resources and seek feasible solutions for future OBI research.

In this work, we propose the first comprehensive OBI-focused benchmark, **OBI-Bench**, to evaluate the recent LMMs in whole-process OBI tasks including recognition, rejoining, classification, retrieval, and deciphering under different settings. Our benchmark is constructed around a key question:

*Can LMMs aid in study of ancient script on oracle bones?*

Specifically, we define two general evaluation principles for LMMs in targeting OBI problems:

- *Task-oriented Perception.* As shown in the bottom of Fig. 1, an LMM is expected to answer accurately to achieve the objectives of five *in-process* tasks like an OBI specialist, such as bounding the position of each oracle bone character for the recognition task or providing a reasonable interpretation of OBIs for the deciphering task.

- *Spanning from excavation to synthesis.* As shown in the middle of Fig. 1, an LMM should be able to handle five different forms of oracle bone inscriptions and show good adaptiveness to the appearance or structural variants of OBI within the same task.

To systematically evaluate the whole-process perception ability on various visual granularity under diverse OBI tasks, we collect 5,523 OBI images from 11 distinct sources. Due to the lack of publicly available OBI recognition datasets on real oracle bones and OBI rejoining datasets, we propose the

Table 1: Overview of the 11 different image source datasets in the **OBI-Bench**, and the respective dataset attributes among **recognition**, **rejoining**, **classification**, **retrieval**, and **deciphering** tasks.

| Task | Image Source Dataset | Type | Sampled Size | Avg. Res. | *Remarks* |
|---|---|---|---|---|---|
| Recognition | YinQiWenYuan$_{detection}$ (AYNU, 2020) | Fragments (inked) | 2,000 | 435×538 | Coordinate |
| | O2BR (Ours) | Fragments (original) | 800 | 1529×1192 | Coordinate |
| Rejoining | OBI-rejoin (Ours) | Fragments (*mixed*) | 483 | 265×313 | Adjacency |
| Classification | HWOBC (Li et al., 2020) | Handprinted | 500 | 400×400 | Class label |
| | Oracle-50K (Han et al., 2020) | Handprinted | 500 | 50×50 | Class label |
| | OBI125 (Yue et al., 2022) | Cropped words | 500 | 77×135 | Class label |
| Retrieval | OBC306 (Huang et al., 2019) | Cropped words | 500 | 68×111 | Class label |
| | OBI-IJDH (Fujikawa & Meng, 2020) | Cropped words | 100 | 60×91 | Class label |
| Deciphering | EVOBC (Guan et al., 2024a) | Handprinted | 50 | 207×212 | Interpretation |
| | OBI Component 20 (Hu et al., 2024) | Handprinted | 50 | 66×64 | Interpretation |
| | HUST-OBS (Wang et al., 2024a) | Handprinted | 40 | 112×172 | Interpretation |

original oracle bone recognition (**O2BR**) dataset and **OBI-rejoin** dataset as complements. Aligned with existing practices (Yue et al., 2024; Wu et al., 2024), each image in OBI-Bench is equipped with a question alongside a correct answer. Moreover, we craft a spectrum of questions: *What* question, *Yes-or-No* question, *How* question, and *Where* question, evaluating from coarse-grained perception to finer-grained perception. A total of **23** popular LMMs are selected for evaluation including 6 proprietary LMMs (*e.g.*, *GPT-4o*, *Qwen-VL-Max*, and *Gemini 1.5 Pro*) and 17 open-source LMMs (*e.g.*, *LLaVA-Next*, *InternVL2*, and *mPLUG-Owl3*). We showcase some empirical findings here: 1) LMMs still have much room for improvement in fine-grained OBI recognition tasks such as quantity detection or locating; 2) LMMs are not sensitive enough to local information such as the borders of the fragmented margin to directly meet the requirements of rejoining. Meanwhile, part of LMMs can help OBI scholars narrow down the range of qualified fragments (GPT-4o and Gemini 1.5 Pro achieve an average performance of 77.57% on the *Acc@10* metric); 3) LMMs can be applied in classification and retrieval tasks, especially for cleaner handprinted datasets; 4) partial LMMs rival the performance of public-level humans in deciphering tasks and exhibit remarkable visual reasoning ability in interpreting those genuine undeciphered oracle bone characters. The full list of our findings from different OBI tasks is in Sec. 3.

By evaluating up-to-date LMMs from different perspectives (OBI research demands), we gain insights into their strengths, limitations, and potential directions for improvement. Ultimately, our objective is to facilitate the field of paleography by fostering the development of more reliable, unbiased, and task-oriented LMMs that meet the needs of OBI experts while upholding trustworthy standards and contributing to historical debates.

## 2 CONSTRUCTING THE OBI-BENCH

### 2.1 GENERAL PRINCIPLES

**Focusing on OBI Task-oriented Abilities of LMMs.** Different from existing LMM benchmarks (Liu et al., 2023b; Li et al., 2024b; Liu et al., 2024b; Yue et al., 2024) that either aim at all-round abilities or generic scenarios, the evaluation principles in **OBI-Bench** are OBI task-oriented. Specifically, we focus on five major issues in the field of oracle bone inscription research: 1) **Recognition**, involves the positioning and identification of OBI characters on different carriers. 2) **Rejoining**, refers to the process of stitching broken OBI fragments together to rebuild complete inscriptions. 3) **Classification**, means categorizing the recognized OBI characters into groups based on their shapes or meanings. 4) **Retrieval**, is to find a cognate character from large collections of OBIs. 5) **Deciphering**, includes interpreting the meaning of the oracle bone characters even the contextual semantic information. We adhere to the principles in designing the visual and cognitive tasks, making the proposed OBI-Bench a focused reflection on the OBI processing abilities of LMMs.

**Covering Multi-stage Font Appearances.** To cover the diverse appearances of the oracle bones since their excavation, we collect multi-sourced images for each task, as depicted in Tab. 1. In particular, considering that there are currently no available original oracle bone recognition datasets and rejoining datasets, we proposed the **O2BR** and **OBI-rejoin** datasets respectively (See more details in App. B). Due to the specific requirement of domain knowledge in OBI research, three domain experts and one senior OBI scholar were involved in the annotation process of O2BR and OBI-rejoin as well as the sampling process of all source datasets. We include all the manifestations of

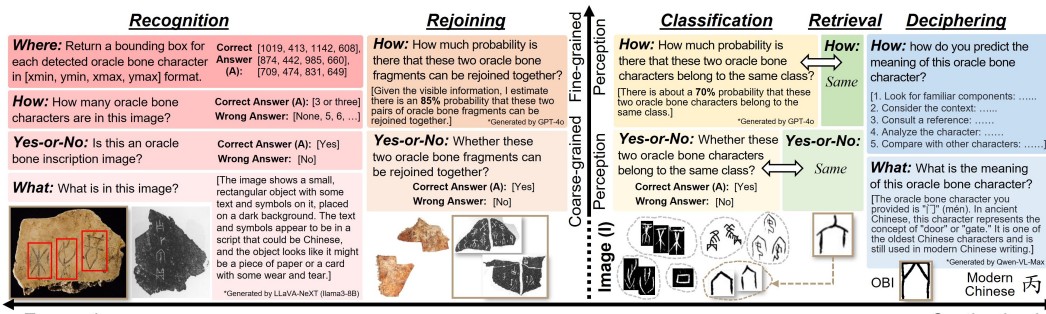

Figure 2: Sampled **OBI-Bench** examples from each task. 5,523 (I,Q,A) tuples span two quadrants of OBI concerns and encompass four types of questions, providing an all-around evaluation of the ability of LMMs on OBI tasks. Note that OBI classification and retrieval tasks share the same queries.

the entire process of OBI processing from excavation (*i.e.*, *original oracle bone*) to artificial synthesis (*i.e.*, *handprinted or computer-generated*). The diverse and multiple sources of images morph the OBI-Bench into a holistic and balanced benchmark to fairly evaluate the competency of LMMs in OBI-related tasks.

## 2.2 BENCHMARK FROM VISUAL ABILITY TO COGNITIVE ABILITY

In the five tasks of OBI-Bench, we evaluate the visual perception and high-level cognitive abilities of LMMs to examine whether they can handle OBI problems by using simple natural queries. For this purpose, we collect 5,523 images (I) from multiple sources (Tab. 1) with diverse task concerns. Then, we design different question prompts (Q) based on the type of tasks and obtain the correct answer (A) from the corresponding dataset for each image. The 5,523 (I, Q, A) tuples compose into the first visual question answering (VQA) testbench (Fig. 2) in the OBI field. Specifically, the questions in OBI-Bench cover two quadrants of concerns for evaluation (Sec. 2.2.1) and four question types (Sec. 2.2.2). The details are elaborated as follows.

### 2.2.1 QUADRANTS FOR EVALUATION CONCERNS

**Axis 1: From Coarse-grained Perception to Fine-grained Perception.** The primary axis differentiates the visual perception granularity of OBI tasks. First, coarse-grained perception focuses on the basic forms, themes, or overall structures of the oracle bone inscriptions, which distinguishes OBIs from other symbols or ancient characters (Qiao et al., 2024). Second, fine-grained perception (Xing et al., 2019; Fujikawa et al., 2023) not only involves identifying subtle characteristics, quantity, and spatial positions of OBIs but also reflects the high-level cognitive ability of LMMs (*e.g.*, *OBI domain expertise* and *interdisciplinary reasoning*). Several studies (Liu et al., 2023a; He et al., 2024; Tong et al., 2024) followed this paradigm and built on it to refine pixel-level perception while extending interactions beyond text-specific inputs.

**Axis 2: From Excavation to Synthesis.** In the era predating the prevalence of artificial intelligence (AI), unearthed oracle bone fragments were normally processed through the manual rubbing and image cropping processes for preservation. After thousands of years of natural weathering, corrosion, and man-made destruction, there are various types of noise in oracle bone rubbings. To improve the readability of the original OBIs and expand the data size that could be processed by deep learning algorithms, researcher obtained pseudo-oracle bone characters through handwriting or generative models (Han et al., 2020; Li et al., 2020; Wang et al., 2024b). Acknowledging these forms of OBIs, we curate 5,523 OBI images from different stages of processing, that require LMMs to grasp the content or other relevant domain knowledge to answer correctly.

### 2.2.2 QUESTION TYPES

In OBI-Bench, we design four question types, *What*, *Yes-or-No*, *How*, and *Where*, to simulate multiple human queries in various OBI tasks, each tailored to different levels of granularity as follows.

• *What* **Question.** The *What* questions are a common type of query in some LMM benchmarks (Wu et al., 2024; Zhang et al., 2024b). In OBI-Bench, they serve as global coarse-grained perception in

Table 2: Performance comparison on the **OBI recognition** task. The best result is marked in **bold** and the second-best result is underlined for both proprietary and open-source LMMs respectively.

| Datasets | O2BR | | | | YinQiWenYuan_detection | | | |
|---|---|---|---|---|---|---|---|---|
| LMM (variant) | What↑ | Yes-or-No↑ | How↓ | Where↑ | What↑ | Yes-or-No↑ | How↓ | Where↑ |
| HUMAN (public) | 0.9364 | 100% | 0.0033 | 0.9272 | 0.9189 | 100% | 0.0060 | 0.8776 |
| **Proprietary LMMs:** | | | | | | | | |
| GEMINI 1.5 PRO | 0.5726 | 98.75% | 0.4932 | **0.1126** | 0.3557 | 99.85% | 0.3811 | **0.0586** |
| GEMINI 1.5 FLASH | 0.4123 | 96.50% | 1.7857 | 0.0962 | 0.3425 | 97.35% | 0.4522 | 0.0276 |
| GPT-4v | 0.5408 | 99.30% | 0.4223 | 0.0022 | 0.3701 | 99.65% | 0.4383 | 0.0165 |
| GPT-4o (ver. 0806) | **0.6114** | **99.95%** | 0.4016 | 0.0038 | **0.3734** | 99.90% | **0.3458** | 0.0182 |
| QWEN-VL-MAX (ver. 0809) | 0.6071 | 99.63% | 0.4799 | 0.0086 | 0.3375 | 99.55% | 0.4843 | 0.0131 |
| GLM-4V | 0.5319 | 39.17% | **0.3681** | 0.0041 | 0.3635 | 52.45% | 0.3632 | 0.0124 |
| **Open-source LMMs:** | | | | | | | | |
| xGen-MM (Instruct-interleave-4B) | 0.5236 | 81.75% | 1.2437 | 0.0233 | 0.3669 | **100%** | 3.4121 | 0.0515 |
| mPLUG-Owl3 (Qwen2-7B) | 0.3342 | **99.88%** | 0.4474 | 0.0811 | 0.2505 | 99.95% | 0.6593 | 0.0527 |
| MiniCPM-V 2.6 (Qwen2-7B) | 0.5576 | 88.75% | 0.4829 | 0.0384 | 0.3781 | 99.10% | 1.1793 | 0.0111 |
| Moondream2 (ver. 0728) | 0.4818 | 98.25% | 0.6795 | 0.0400 | 0.3049 | 91.30% | 0.4436 | **0.0653** |
| InternVL2-Llama3-76B (Llama3-70B) | **0.5833** | 99.65% | **0.4328** | 0.0976 | 0.3892 | 99.75% | 0.5344 | 0.0623 |
| InternVL2-40B (Nous-Hermes2-Yi-34B) | 0.5664 | 98.35% | 0.4561 | 0.0766 | 0.3637 | 99.05% | 0.6733 | 0.0487 |
| InternVL2-8B (InternLM2.5-7B) | 0.5232 | 95.00% | 0.4618 | 0.0020 | 0.3429 | 97.95% | 1.1146 | 0.0152 |
| GLM-4V-9B (GLM-4-9B) | 0.5388 | 29.50% | 0.4825 | <10e-4 | 0.2839 | 15.70% | **0.3934** | <10e-4 |
| CogVLM2-Llama3-19B (Llama3-8B) | 0.5321 | 61.00% | 0.6928 | <10e-4 | **0.3966** | 91.75% | 0.5568 | 0.0002 |
| LLaVA-NeXT (Qwen1.5-72B) | 0.4846 | 99.75% | 1.1011 | 0.0445 | 0.3297 | 99.75% | 1.0561 | 0.0591 |
| LLaVA-NeXT (Llama3-8B) | 0.4764 | 93.13% | 0.5512 | 0.0001 | 0.3120 | 93.90% | 0.4268 | 0.0189 |
| IDEFICS-2-8B (Mistral-7B) | 0.3175 | 95.88% | 0.4916 | <10e-4 | 0.2658 | 95.10% | 0.5119 | <10e-4 |
| DeepSeek-VL (DeepSeek-LLM-7B) | 0.5111 | 92.75% | 0.5657 | 0.0449 | 0.3386 | 98.00% | 0.6263 | 0.0520 |
| InternLM-XComposer2-VL (InternLM2-7B) | 0.5106 | 99.88% | 0.6641 | 0.0049 | 0.2661 | 99.95% | 1.5231 | 0.0281 |
| LLaVA-v1.5 (Vicuna-v1.5-13B) | 0.4416 | 99.00% | 2.7553 | 0.0751 | 0.2875 | 98.05% | 1.7662 | 0.0493 |
| LLaVA-v1.5 (Vicuna-v1.5-7B) | 0.4239 | 91.88% | 0.8861 | 0.0656 | 0.2729 | 81.90% | 1.9621 | 0.0465 |
| Qwen-VL (Qwen-7B) | 0.4489 | 74.13% | 1.7416 | 0.0003 | 0.3137 | 86.81% | 3.5694 | 0.0069 |

recognition task (*e.g., What is in this image?*), or unbound the answers in deciphering task (*e.g., What is the meaning of this oracle bone character?*). In contrast to other question types, the *What* questions examine more comprehensive visual understanding and cognitive abilities of LMMs, by requiring correct attribute perception and knowledge integration.

• *Yes-or-No* **Question.** As a fundamental type of judgment, *Yes-or-No* represents a binary output. This design minimizes the ambiguity of the model answers and is closer to the task objectives than the open-ended responses of *What* questions.

• *How* **Question.** Except from two common types, we also include the *How* questions to further refine the responses as an extension to *Yes-or-No* questions. As shown in Fig. 2, we can query *How many oracle bone characters are in this image?* for the recognition task (Fu et al., 2023) or query *How much probability is there that these two oracle bone characters belong to the same class?* for the classification task to derive quantitative results predicted by LMMs.

• *Where* **Question.** We specifically employ the *Where* question for the recognition task. For instance, we can query *Return a bounding box for each detected oracle bone character in [xmin, ymin, xmax, ymax] format* to achieve pixel-level OBI anchoring, which requires the LMM to have a fine-grained perception, including spatial location, shape, and relationships between detected objects.

## 3 EXPERIMENTS

In OBI-Bench, we evaluate **17** up-to-date popular and competitive open-source LMMs, together with **6** proprietary LMMs, under zero-shot settings. More results and analyses are appended in App. C.

### 3.1 EVALUATION ON RECOGNITION

In this section, we examine the recognition ability of LMMs, comparing them from coarse-grained global content perception to fine-grained OBI locating.

**Setup.** To evaluate how well LMMs perform under various OBI recognition scenarios, we design four evaluation schemes across two image sources (*i.e.*, original oracle bone from our self-curated O2BR dataset and inked rubbings from the YinQiWenYuan_detection (AYNU, 2020) dataset): (**1**) evaluating via the *What* query to measure the preciseness of the global perceptual descriptions on OBIs. Given the commonality of contents in sampled test sets, we predefined reference descriptions for each of the

Table 3: Results on the **OBI rejoining** task. *'Yes-or-No'* and *'How'* represent the absolute output and probability output, respectively. We report *Acc@5* for the *'Yes-or-No'* query. Given that not all LMMs support multi-image inputs natively, we shrink the number of evaluated models.

| **LMM** (*variant*) | **Yes-or-No**↑ | **How**↑ | | |
|---|---|---|---|---|
| | | *Acc@1* | *Acc@5* | *Acc@10* |
| GEMINI 1.5 PRO | 28.53% | 24.88% | 46.37% | 76.67% |
| GEMINI 1.5 FLASH | 22.17% | 19.33% | 36.34% | 66.76% |
| GPT-4V | 27.63% | 26.86% | 43.75% | 73.75% |
| GPT-4O (*ver. 0806*) | **32.21%** | **29.13%** | **48.43%** | **78.47%** |
| QWEN-VL-MAX (*ver. 0809*) | 21.77% | 13.14% | 31.68% | 56.67% |
| GLM-4V | 5.33% | 4.58% | 12.19% | 21.46% |
| xGen-MM (*Instruct-interleave-4B*) | 12.08% | 3.11% | 9.32% | 11.18% |
| mPLUG-Owl3 (*Qwen2-7B*) | 14.48% | 3.52% | 12.63% | 17.60% |
| MiniCPM-V 2.6 (*Qwen2-7B*) | 13.23% | 2.69% | 11.18% | 16.98% |
| InternVL2-Llama3-76B (*Llama3-70B*) | **20.13%** | **5.18%** | **16.98%** | **31.68%** |
| InternVL2-40B (*Nous-Hermes2-Yi-34B*) | 16.00% | 3.31% | 10.97% | 24.84% |
| InternVL2-8B (*InternLM2.5-7B*) | 14.68% | 3.73% | 10.14% | 17.81% |
| LLaVA-NeXT (*Qwen1.5-72B*) | 17.45% | 4.97% | 14.29% | 27.74% |
| LLaVA-NeXT (*Llama3-8B*) | 11.41% | 3.93% | 9.52% | 16.77% |
| IDEFICS-2-8B (*Mistral-7B*) | 7.38% | 3.52% | 12.42% | 14.91% |
| DeepSeek-VL (*DeepSeek-LLM-7B*) | 9.40% | 4.35% | 8.90% | 11.59% |
| Qwen-VL (*Qwen-7B*) | 9.86% | 4.02% | 8.96% | 14.08% |

two datasets. The `max_tokens` for all LMMs is set to 200. The similarity between text embeddings is used as the evaluation metric. (**2**) evaluating via the *Yes-or-No* query after specifying the response direction (*the words "OBI" appear explicitly in the query*), designed to complement the evaluation on coarse-grained perception. The accuracy of answers is adopted as the evaluation criterion. (**3**) evaluating via the *How* query to quantify the ability of LMMs in parsing irregularly structured character-level OBI contents. We use mean relative error (MRE) as the metric. (**4**) evaluating via the *Where* query to achieve finer-grained perception (*i.e.*, dense oracle bone character locating), designed to more effectively uncover the practicability of LMMs in real OBI processing pipelines. Mean intersection over union (mIoU) is used as the metric. More details are in App. C.2.

**Results.** In scenario (**1**), as shown in Tab. 2, among proprietary LMMs, the recently-released GPT-4o reaches the best performance in terms of the relevance of answers on coarse-grained perception, followed by Qwen-VL-Max, which shows rather close results. By achieving over 12.4% on average performance improvement, these models provide more precise answers than the open-source LMMs. Similar improvement can be observed for the same series of LMM (*e.g.*, *LLaVA* and *InternVL2*), as the number of parameters of the language backbone increases. In scenario (**2**), we add keyword (*i.e.*, *oracle bone*) directly to a *Yes-or-No* question to further evaluate the global perception of LMMs. We find most LMMs perform nearly 100% correctness, indicating that these models are highly effective in handling queries that involve explicitly directed content. One exception is the GLM-4V series that only achieves an average accuracy of 34.34%. We speculate it is due to the answer preference bias (Wu et al., 2024). A deeper analysis is in App. C.2.2. In scenario (**3**), we transition to examining the fine-grained visual perceptual of LMMs, *i.e.*, counting the number of oracle bone characters. Our results in Tab. 2 indicate that GLM-4V and InternVL2-Llama3-76B exhibit the best performance within their respective division groups. Additionally, we observe some abnormally large values (>50) or repeated values from Gemini 1.5 Flash, xGen-MM-4B, LLaVA-NeXT-72B, LLaVA-v1.5-13B, and Qwen-VL that result in their relatively large MRE (Zhou et al., 2024). In scenario (**4**), all evaluated LMMs consistently underperform. Gemini 1.5 Pro significantly outperforms other LMMs (*surpasses GPT-4v and GPT-4o two orders of magnitude*) (Ueno & Lynn, 2024) with its exclusive coordinate return function. Overall, current LMMs are still not usable for character-level OBI locating and are far from the *public-level human*.

## 3.2 EVALUATION ON REJOINING

In this section, we delve into the rejoining performance of LMMs against various eroded, tiny, and irregularly structured fragments of OBI, focusing on the estimation of rejoinable fragments. This task is also a major concern in the recovery of paper money, calligraphy, and painting files recovery.

**Setup.** Given the absence of publicly available rejoining datasets in the OBI domain, we create a dataset consisting of 200 complete OBI pieces across two appearances (*i.e.*, original and inked

Table 4: Classification accuracy (%) on HWOBC, Oracle-50k, and OBI125 datasets (delimited by slashes) under different queries. *'Yes-or-No'* and *'How'* represent the absolute output and probability output, respectively. We report *Acc@5* for the *'Yes-or-No'* query.

| LMM (*variant*) | Yes-or-No↑ | How↑ Acc@1 | How↑ Acc@5 |
|---|---|---|---|
| GEMINI 1.5 PRO | 68.75%/68.00%/58.50% | 88.75%/85.50%/**77.50%** | 100.0%/100.0%/93.75% |
| GEMINI 1.5 FLASH | 62.50%/60.50%/50.25% | 82.00%/82.50%/71.75% | 100.0%/99.50%/91.00% |
| GPT-4v | 69.50%/66.00%/57.75% | 86.75%/88.25%/70.75% | 100.0%/100.0%/91.75% |
| GPT-4o (*ver. 0806*) | **72.75%/74.50%/62.50%** | **89.75%/90.25%**/75.50% | 100.0%/100.0%/93.50% |
| QWEN-VL-MAX (*ver. 0809*) | 64.25%/65.75%/55.00% | 85.00%/88.75%/69.25% | 100.0%/98.75%/89.75% |
| GLM-4V | 35.50%/29.75%/34.50% | 71.00%/68.50%/54.25% | 83.50%/85.75%/77.75% |
| xGen-MM (*Instruct-interleave-4B*) | 36.25%/37.75%/35.75% | 46.75%/48.00%/43.50% | 57.75%/55.00%/49.00% |
| mPLUG-Owl3 (*Qwen2-7B*) | 39.25%/39.75%/38.75% | 44.75%/46.25%/43.75% | 56.25%/52.50%/53.50% |
| MiniCPM-V 2.6 (*Qwen2-7B*) | 40.75%/42.50%/38.50% | 45.75%/45.00%/42.75% | 56.75%/55.75%/51.75% |
| InternVL2-Llama3-76B (*Llama3-70B*) | **44.75%**/47.50%/43.25% | **53.75%/55.00%/50.75%** | **69.75%/69.00%/66.75%** |
| InternVL2-40B (*Nous-Hermes2-Yi-34B*) | 42.75%/43.50%/40.25% | 49.75%/50.00%/48.75% | 63.75%/62.25%/59.75% |
| InternVL2-8B (*InternLM2.5-7B*) | 42.25%/41.75%/38.75% | 47.75%/49.00%/47.75% | 59.75%/59.00%/56.50% |
| LLaVA-NeXT (*Qwen1.5-72B*) | 46.25%/45.50%/41.00% | 51.75%/52.00%/49.00% | 66.75%/67.75%/64.25% |
| LLaVA-NeXT (*Llama3-8B*) | 44.00%/42.00%/38.75% | 46.75%/46.25%/42.75% | 53.75%/56.75%/54.75% |
| IDEFICS-2-8B (*Mistral-7B*) | **44.75%**/44.50%/39.75% | 46.75%/46.50%/44.00% | 58.75%/58.75%/53.75% |
| DeepSeek-VL (*DeepSeek-LLM-7B*) | 42.25%/40.75%/35.75% | 47.00%/47.25%/45.25% | 56.75%/58.25%/58.50% |
| Qwen-VL (*Qwen-7B*) | 44.25%/42.00%/38.75% | 48.00%/51.00%/44.75% | 61.25%/63.50%/61.25% |

oracle bone fragments) with 483 rejoinable fragments. More details are in App. B.2. To evaluate the rejoining performance of LMMs on oracle bone fragments with different marginal sealability concerns, we elaborate two evaluation *scenarios*: (**1**) evaluating via standard *Yes-or-No* question that represents the absolute judgments to tentatively explore the rejoining performance; (**2**) evaluating via *How* question that outputs rejoinable probabilities for pairs of fragments, allowing for a more accurate and quantitative measurement. *Accuracy@k* indices are utilized as evaluation metrics.

**Results.** Tab. 3 presents the evaluation results of LMMs on the OBI rejoining task, from which we gain several valuable observations as follows. First, GPT-4o and InternVL2-Llama3-76B achieve the best performance in their respective categories, although there is still a huge gap between them. Second, among all proprietary LMMs, the rejoining performance under *'How'* queries is significantly better than that under *'Yes-or-No'* queries. This indicates that the answer in probabilistic form better reflects the differences in visual perception compared to absolute form output. Third, we notice that GPT-4o and Gemini 1.5 Pro reach over 76% in *Acc@10* metric, illustrating the possibility of of using LMMs to assist the OBI rejoining efforts. Fourth, the number of parameters is roughly proportional to the performance, and newer models tend to exhibit superiority under the same scale of language backbone models. Fifth, we find that the overall performance of open-source LMMs is still far away from being truly usable (only 3.85%, 11.39%, and 18.65% on average in terms of *Acc@1*, *Acc@5*, and *Acc@10*, respectively), and their ranks vary greatly under different evaluation criteria.

## 3.3 EVALUATION ON CLASSIFICATION

In this section, we evaluate the performance of LMMs on OBI classification tasks and try to answer: *(1) How robust are LMMs in classifying oracle bone characters?* Since the intra-class and inter-class similarity of OBIs are sometimes numerically close, we then investigate the capabilities of LMMs in focusing on atypical characteristics and try to answer: *(2) How well do LMMs distinguish between different categories of OBIs?*

**Setup.** We sample 500 images across 100 categories from each of the HWOBC, Oracle-50k, and OBI125 datasets as test set. One image is kept as the label in each category and the remaining four as query images. To answer the above questions, we construct three evaluation *scenarios*: (**1**) evaluation on oracle bone rubbings and handprinted images, aiming to assess the robustness of LMMs to content qualities. Due to the inadequacy of manual rubbing and the limitation of scanning equipment, there are inevitably various noises in rubbing images, which could greatly affect the classification accuracy; (**2**) evaluation on different queries (*i.e.*, *Yes-or-No* question and *How* question) designed to generate absolute and probability forms of responses, aiming to assess the robustness of LMMs under varied query prompts; (**3**) evaluation on {10, 20, 30, 50, 100} classes that encompass mixed image sources and subsets characterized by similar structures, aiming to study the conditions under which LMMs will fail in classification. We thereby reconstruct a variable-quantity mixed test set by sampling an equal number of oracle bone character classes from three datasets. More details are in App. C.3.

Table 5: Performance comparison of LMMs on the OBI retrieval task. Results on OBC306 and OBI-IJDH datasets are delimited by slashes. We report the averaged *Recall@1*, *Recall@3*, and *Recall@10* on 'How' question.

| LMM (*variant*) | Yes-or-No↑ | How↑ | | | |
|---|---|---|---|---|---|
| | $mAP@\|Yes\|$ | Recall@1 | Recall@3 | Recall@10 | mAP@5 |
| GEMINI 1.5 PRO | 0.4316/**0.5734** | 0.225/**0.250** | 0.671/0.688 | **1.000/1.000** | 0.644/0.76 |
| GEMINI 1.5 FLASH | 0.3876/0.5344 | 0.195/0.225 | 0.635/0.644 | 0.975/**1.000** | 0.562/0.74 |
| GPT-4v | 0.4228/0.5680 | 0.205/0.225 | 0.650/0.676 | **1.000/1.000** | 0.624/0.74 |
| GPT-4O (*ver. 0806*) | **0.4550/0.6122** | **0.235/0.250** | 0.686/0.706 | **1.000/1.000** | **0.688/0.80** |
| QWEN-VL-MAX (*ver. 0809*) | 0.4223/0.5716 | 0.190/0.225 | 0.621/0.638 | **1.000/1.000** | 0.664/0.78 |
| GLM-4V | 0.3018/0.3867 | 0.125/0.175 | 0.495/0.613 | 0.925/**1.000** | 0.520/0.72 |
| xGen-MM (*Instruct-interleave-4B*) | 0.2668/0.3423 | 0.085/0.200 | 0.336/0.569 | 0.855/0.975 | 0.366/0.66 |
| mPLUG-Owl3 (*Qwen2-7B*) | 0.2814/0.3550 | 0.075/0.225 | 0.371/0.588 | 0.870/0.950 | 0.384/0.62 |
| MiniCPM-V 2.6 (*Qwen2-7B*) | 0.2863/0.3575 | 0.070/0.225 | 0.361/0.581 | 0.885/0.925 | 0.368/0.66 |
| InternVL2-Llama3-76B (*Llama3-70B*) | **0.3557/0.4268** | 0.150/**0.250** | 0.460/0.675 | **1.000/1.000** | 0.522/**0.72** |
| InternVL2-40B (*Nous-Hermes2-Yi-34B*) | 0.3260/0.3914 | 0.125/0.250 | 0.445/0.656 | **1.000/1.000** | 0.480/0.68 |
| InternVL2-8B (*InternLM2.5-7B*) | 0.2844/0.3623 | 0.095/0.225 | 0.374/0.650 | 0.925/**1.000** | 0.420/0.68 |
| LLaVA-NeXT (*Qwen1.5-72B*) | 0.3478/0.4143 | **0.155/0.250** | 0.468/0.669 | **1.000/1.000** | **0.562/0.70** |
| LLaVA-NeXT (*Llama3-8B*) | 0.2793/0.3605 | 0.075/0.225 | 0.358/0.606 | 0.925/**1.000** | 0.348/0.60 |
| IDEFICS-2-8B (*Mistral-7B*) | 0.2844/0.3602 | 0.090/0.200 | 0.354/0.619 | 0.875/**1.000** | 0.402/0.66 |
| DeepSeek-VL (*DeepSeek-LLM-7B*) | 0.2867/0.3450 | 0.105/0.225 | 0.343/0.600 | 0.930/**1.000** | 0.416/0.64 |
| Qwen-VL (*Qwen-7B*) | 0.2883/0.3528 | 0.080/0.225 | 0.345/0.588 | 0.920/0.925 | 0.422/0.66 |

**Results.** In scenario (**1**), from Tab. 4, we can observe that the classification results on HWOBC and Oracle-50k datasets are markedly better than those on the OBI125 dataset (surpass by 16.4% and 19.2% for GPT-4o under 'Yes-or-No' queries). We believe that this can be attributed to the image quality of OBI in the dataset, which affects the LMM's perception of character structure. Indeed, the two handprinted OBI datasets, HWOBC and Oracle-50k, are cleaner than the noise-contaminated OBI125 dataset (Sec. C.3). In scenario (**2**), we find significant performance discrepancies between 'Yes-or-No' and 'How' queries, which is similar to the other tasks. This highlights the necessity of designing more appropriate textual answer forms and task-specific metrics. Moreover, the Gemini 1.5 series is

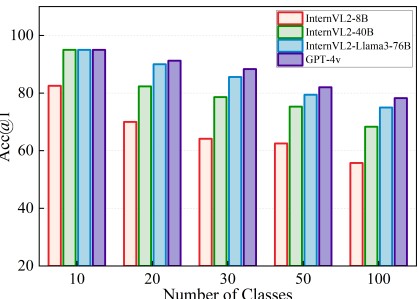

Figure 3: Effects of the number of character categories on classification accuracy.

on par with that of their best competitors (GPT-4v and GPT-4o), which even achieves nearly 90% and 100% accuracy in terms of *Acc@1* and *Acc@5* metrics respectively. This proves the feasibility of using LMMs for preliminary classification of OBI images. In scenario (**3**), we select four representative LMMs, *i.e.*, InternVL2-8B, InternVL2-40B, InternVL2-Llama3-76B, and GPT-4v, for evaluation, and obtain three observations from Fig. 3. First, we notice an overall performance improvement compared to the previous experimental setup since the source-mixed test set increases the inter-class differences. Second, the accuracy of these models decreases as the number of classes increases. This shows that current LMMs struggle to perform structural comparisons across a large number of categories, potentially limited by the granularity of their outputs. Third, compared to models with smaller scales, the performance degradation of InternVL2-Llama3-76B and GPT-4v is less pronounced as the number of classification categories increases, indicating that LMMs with larger parameter sizes tend to have better robustness.

## 3.4 EVALUATION ON RETRIEVAL

In addition to classification, the retrieval task is also critical for building large-scale OBI image databases. Considering that LMMs combine powerful visual perception and natural language understanding capabilities, they can serve as the demand processing core of OBI retrieval systems. Here, we evaluate the performance of LMMs on OBI retrieval tasks given different types of queries.

**Setup.** We adopt two query formats (*i.e.*, Yes-or-No and How questions) akin to those employed in the classification tasks. A total of 600 images sampled from OBC306 and OBI-IJDH datasets are used for evaluation. Among each test set, half of the images are used as distractors to simulate a real OBI retrieval scenario. During experiments, OBI images in the same category are retrieved. We take the averaged *Recall@k* and mean Average Precision (mAP) metrics to quantify the OBI retrieval

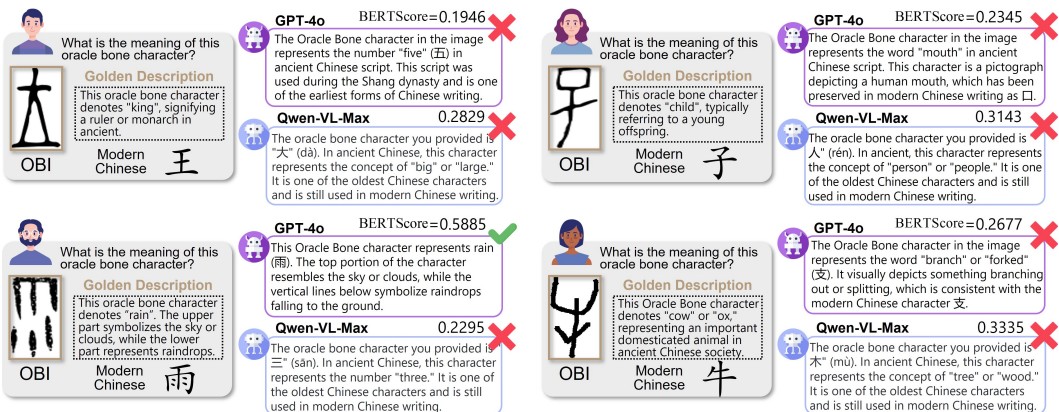

Figure 4: Qualitative comparison of deciphering results between two *state-of-the-art* LMMs, *i.e.*, GPT-4o and Qwen-VL-Max, in a single round of direct questioning. It is noted that neither GPT-4o nor Qwen-VL-Max has fully deciphered these four oracle bone characters.

performance of LMMs in multi-round queries. Note that we use *mAP@|Yes|* as the evaluation metric in *Yes-or-No* scenarios, where $|Yes|$ denotes the number of '*Yes*' answers.

**Results.** As presented in Tab. 5, most proprietary LMMs can well retrieve relevant OBI contents from test pools except GLM-4v, which is on average 40.48% worse than the other five in terms of *Recall@1*. Besides, when expanding the search scope to ten candidates, all proprietary LMMs as well as over 63.6% open-source LMMs perfectly retrieve all relevant OBI contents in OBI-IJDH. However, the retrieval performance fell off a cliff in the OBC306 dataset, which we attribute to the same image quality problems encountered with the classification task. As visualized in Fig. 1, such cropped oracle bone rubbings contain various distortion problems, such as stroke-broken, edge-cracked, spindles, and dense white regions, making them harder to tell apart and showing great necessity to add image enhancement measures before these tasks. Furthermore, models with a large number of parameters still perform relatively better in this task and *How* queries receive more precise search results.

## 3.5 EVALUATION ON DECIPHERING

Deciphering the oracle bone inscriptions has long been the ultimate goal of this ancient language study. Large multi-modal models have strong cross-modal understanding capabilities, enabling the models to perform multilingual visual-text transformation with linguistic analysis. In this section, we examine the potential OBI deciphering abilities of LMMs by asking: *(1) Can LMMs interpret oracle bone characters correctly? (2) How do models behave in the face of different character formation principles (e.g., pictograph, ideogram, and radical-based variants) and character frequencies? (3) What is the reasoning mechanism of LMMs during the OBI deciphering process?*

**Setup.** To evaluate the decipherment performance of LMMs, we construct four evaluation *scenarios*: (**1**) deciphering oracle bone characters with different frequency levels $F$. Three test sets (*i.e.*, *Tier-1:* $F \geq 500$, *Tier-2:* $100 \leq F \leq 500$, *Tier-3:* $10 \leq F \leq 100$) with a total of 40 characters sampled from the HUST-OBS dataset according to the taxonomy in (Liu, 2010; Chen, 2010); (**2**) deciphering oracle bone characters in *pictograph* formation. We follow the category definitions in (Chou, 1979; Da WEIH, 2021) and choose 28 representative OBI characters from the EVOBC dataset as test set; (**3**) deciphering oracle bone characters in *ideogram* formation. Similar to scenario (**2**), 22 OBI characters from the EVOBC dataset are selected; (**4**) deciphering oracle bone characters with structurally similar variants (*i.e.*, *radical*). We pick out 10 different radicals from the OBI Component 20 dataset (Hu et al., 2024), each accompanied by 5 component-level variants. Fig. 9, Fig. 10, and Fig. 11 in the appendix visualize the sampled test sets for the above scenarios. Different from other approaches (Chang et al., 2022; Guan et al., 2024b) that take the decipherment of OBI as establishing correspondences with modern Chinese characters, we directly use the LMMs to generate a segment of interpretation, as illustrated in Fig. 4. Given the absence of dedicated evaluation criteria for OBI decipherment, we produce text embeddings for all decipherment results and calculate the BERTScore (Zhang et al., 2019) with expert-labelled *golden* descriptions (*i.e.*, *text references*). More details are in App. B.3.

**Results.** For scenario (**1**), Tab. 6 presents the evaluation results across different character frequencies. We find that the evaluated LMMs achieve on average 0.1905, 0.1898, and 0.1791 on *Tier1*, *Tier2*,

Table 6: Results on **OBI deciphering** task. Given a reference description $x = \langle x_1, \ldots, x_k \rangle$ and a candidate deciphered result $\hat{x} = \langle \hat{x_1}, \ldots, \hat{x_k} \rangle$, we use contextual embeddings to represent the tokens, and compute matching using cosine similarity.

| Datasets | HUST-OBS | | | EVOBC | | OBI Component 20 | Average |
|---|---|---|---|---|---|---|---|
| LMM (*variant*) | *Tier-1* | *Tier-2* | *Tier-3* | *pictograph* | *ideogram* | *Radical* | |
| HUMAN (*public*) | 0.4507 | 0.3884 | 0.3437 | 0.4966 | 0.3627 | 0.2812 | 0.3872 |
| GEMINI 1.5 PRO | 0.3766 | **0.3834** | 0.3589 | 0.4226 | 0.3696 | 0.3750 | 0.3810 |
| GEMINI 1.5 FLASH | 0.3545 | 0.3829 | 0.3567 | 0.3661 | 0.3595 | 0.3648 | 0.3641 |
| GPT-4V | 0.3764 | 0.3808 | 0.3596 | 0.4424 | 0.3551 | **0.4060** | 0.3867 |
| GPT-4O (*ver. 0806*) | **0.3891** | 0.3510 | 0.3660 | **0.4535** | **0.3893** | 0.3767 | **0.3876** |
| QWEN-VL-MAX (*ver. 0809*) | 0.2345 | 0.2273 | 0.2322 | 0.2565 | 0.2533 | 0.2785 | 0.2471 |
| GLM-4V | 0.2180 | 0.1638 | 0.2122 | 0.2353 | 0.2451 | 0.1683 | 0.2071 |
| xGen-MM (*Instruct-interleave-4B*) | 0.1174 | 0.1332 | 0.1369 | 0.1526 | 0.1268 | 0.1031 | 0.1283 |
| mPLUG-Owl3 (*Qwen2-7B*) | 0.1609 | 0.1822 | 0.1897 | 0.2036 | 0.1751 | 0.0684 | 0.1633 |
| MiniCPM-V 2.6 (*Qwen2-7B*) | 0.1411 | 0.1694 | 0.1584 | 0.1703 | 0.1529 | 0.0907 | 0.1471 |
| Moondream2 (*ver. 0728*) | 0.1129 | 0.1171 | 0.1063 | 0.1297 | 0.1176 | 0.1155 | 0.1165 |
| InternVL2-Llama3-76B (*Llama3-70B*) | 0.2324 | 0.2139 | 0.2156 | 0.2355 | 0.2123 | **0.1881** | 0.2163 |
| InternVL2-40B (*Nous-Hermes2-Yi-34B*) | 0.1966 | 0.1917 | 0.1883 | 0.2007 | 0.1902 | 0.1376 | 0.1842 |
| InternVL2-8B (*InternLM2.5-7B*) | 0.1676 | 0.1739 | 0.1340 | 0.1746 | 0.1517 | 0.1670 | 0.1615 |
| GLM-4V-9B (*GLM-4-9B*) | 0.1291 | 0.0821 | 0.0478 | 0.0835 | 0.0909 | 0.0399 | 0.0789 |
| CogVLM2-Llama3-19B (*Llama3-8B*) | 0.1096 | 0.0873 | 0.0852 | 0.1024 | 0.1079 | 0.1192 | 0.1019 |
| LLaVA-NeXT (*Qwen1.5-72B*) | 0.1501 | 0.1647 | 0.1372 | 0.1468 | 0.1123 | 0.0863 | 0.1329 |
| LLaVA-NeXT (*Llama3-8B*) | 0.0893 | 0.0873 | 0.0892 | 0.0806 | 0.0813 | 0.0298 | 0.0763 |
| IDEFICS-2-8B (*Mistral-7B*) | 0.1639 | 0.1252 | 0.0985 | 0.1601 | **0.2189** | 0.1517 | 0.1531 |
| DeepSeek-VL (*DeepSeek-LLM-7B*) | 0.0458 | 0.0628 | 0.0504 | 0.0324 | 0.0593 | 0.0931 | 0.0573 |
| InternLM-XComposer2-VL (*InternLM2-7B*) | **0.2963** | **0.3101** | **0.2660** | **0.2650** | 0.1822 | 0.0626 | **0.2304** |
| LLaVA-v1.5 (*Vicuna-v1.5-13B*) | 0.1368 | 0.1669 | 0.1629 | 0.1128 | 0.1214 | 0.1168 | 0.1363 |
| LLaVA-v1.5 (*Vicuna-v1.5-7B*) | 0.0677 | 0.1165 | 0.0906 | 0.0924 | 0.0887 | 0.1488 | 0.1008 |
| Qwen-VL (*Qwen-7B*) | 0.1145 | 0.0912 | 0.0757 | 0.1080 | 0.0635 | 0.0744 | 0.0879 |

and *Tier3* subsets, respectively, indicating that the deciphering performance on common characters is better than that on rare characters. For scenario (**2**) and (**3**), the deciphering results on *pictograph* are more accurate than *ideogram* (average BERTScore: 0.2012 > 0.1837), which shows the characteristics of pictographs evolving from drawings and reveals the deciphering mechanism of LMMs, that is, by establishing visual associations between font structure and real objects. For scenario (**4**), we notice the worst performance (average BERTScore = 0.1636) on the *Radical* subset compared to other test sets, which means LMMs are impeded in distinguishing and deciphering component-level variants. Furthermore, proprietary LMMs outperform open-source LMMs by an average of 145.9% across all deciphering tasks, suggesting that the open-source LMMs still have a substantial gap to fill for practical application. It is worth noting that GPT-4o, GPT-4v, and Gemini 1.5 Pro have approached or exceeded *public-level* humans in some scenarios. Surprisingly, InterLM-XComposer2-VL reaches the *Top-1* performance among all open-source LMMs (even surpassing InternVL2-Llama3-76B and InternVL2-40B which owns several times of parameters). We suspect that this might be attributed to the specific training datasets (Cai et al., 2024) since the match of data content sometimes outweighs the sheer quantity of parameters (Zhang et al., 2024a). Meanwhile, we observe some highly repetitive outputs (such as '*The image you've provided appears to be a stylized representation of an oracle bone character.*' or other meaningless answers) in the LLaVA-NeXT series and its previous version LLaVA-v1.5, which leads to inferior deciphering results in all scenarios. Fig. 4 provides an example of the deciphering results of GPT-4o and Qwen-VL-Max on four high-frequency oracle bone characters. We also investigate the effect of adding pre-prompt during querying in App. C.4.1. More deciphered results for currently undeciphered OBI are presented in App. C.4.2.

## 4 CONCLUSION

We provide comprehensive evaluations of the applicability of LMMs in OBI domain from different perspectives. Overall, there is still considerable room for LMMs to improve when addressing tasks that need domain-specific knowledge. However, taking LMM as a unique technique to assist studies on oracle bone inscriptions is a promising direction, which may significantly reduce learning costs for domain professionals and help them solve OBI-related problems more easily than learning or training traditional deep learning algorithms. Additionally, there are many properties of inputs that affect the performance of OBI processing based on our evaluations, which is worth further exploring. We hope these results can bring insights into the future improvements of LMMs for finer-grained and more robust visual perception. We address the ethical concerns of our research in App. A and discuss the limitations of this work in App. D.

ACKNOWLEDGMENTS

This work was supported in part by the National Key R&D Program of China under Grant 2021YFE0206700, in part by the National Social Science Fund of China under Grant 24Z300404220, and in part by the Shanghai Office of Philosophy and Social Science under Grant 2023BYY003.

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

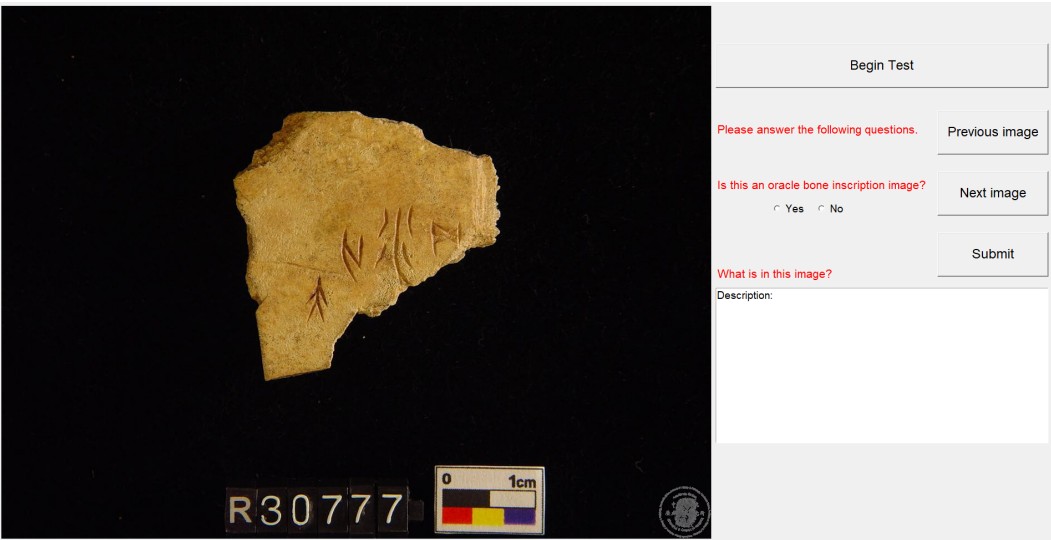

Figure 5: Interface of subjective experiments for the recognition task.

# APPENDIX

# A  ETHICAL DISCUSSIONS

## A.1  ETHICAL DISCUSSIONS OF OUR RESEARCH

Our work holds the potential for significant social impact, both positive and negative. **Firstly**, the main motivation of this work is that the archaeological discoveries of oracle bone inscriptions (OBI) are normally considered as highly serendipitous findings while their interpretation relies heavily on the large volume of manual efforts (*from excavation to digital archiving*), yet most of their meanings are still unknown, hindering historians and sociologists to further understand the contents of production and life in ancient times. In the early stage of OBI research, the deciphering of oracle bone characters is highly limited, either by referring to ancient books or by working backward from the perspective of font evolution, which was in a fairly stagnant state for much time. Recently, with a large volume of large multi-modal models (LMMs) claiming their powerful cross-modal information processing capability, it is unknown for both computer vision experts and OBI scholars if LMMs can identify and decipher them. In our work, we find while most LMMs are still not completely applicable in some OBI processing tasks that need fine-grained perception and deeper cognition, it exhibits a trend that the performance of the model improves year by year and relies on its design for specific tasks. Our findings offer a promising direction that using LMMs to tackle the OBI problems has a lower knowledge cost than the traditional model-task in one-to-one form on certain occasions. **Secondly**, our research on the model applicability in OBI tasks provides a necessary understanding of the nature and potential causes of the inferior perception and cognition abilities of LMMs in the knowledge-specific domain. The evaluation of model performance and the subsequent discoveries would spark a broader discussion on the rational use of LMMs to facilitate the study of ancient characters. Our work could serve as a reference point for discussions on developing next-generation LMMs and evaluation metrics for the OBI area. **Thirdly**, we acknowledge that unregulated and unlimited deciphering results of LMMs may present potential risks in generating misinformation and spreading false cultural values.

## A.2  ETHICAL DISCUSSIONS OF DATA COLLECTION

In this section, we detail the ethical concerns that might arise during the experiments. Specifically, we invite five Chinese college students with no background in the ancient Chinese language to represent public-level performance. They are instructed to perform two tasks that are of great interest to OBI scholars, *i.e.*, recognition and deciphering. Note that all participants are clearly informed of the

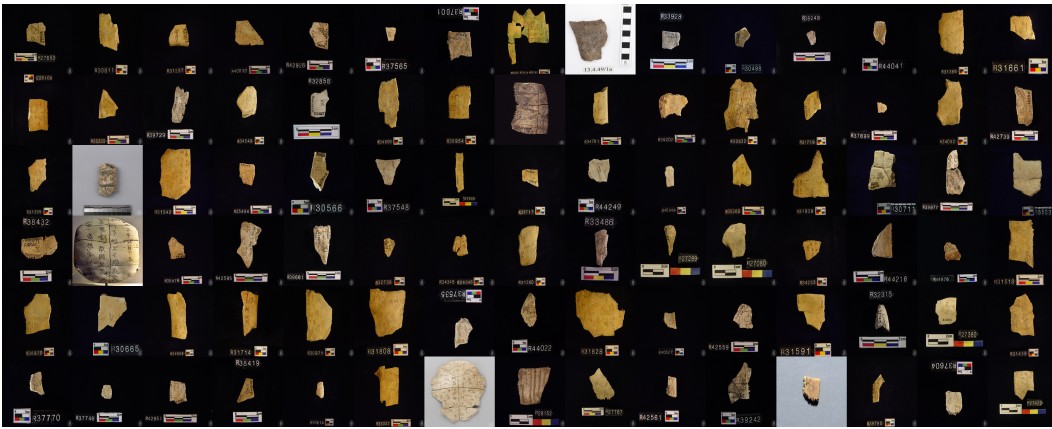

Figure 6: Sample images of the original oracle bone fragments in the O2BR dataset.

contents in our experiments. Similar to (Chen et al., 2024b;c;d), we addressed the ethical challenges by obtaining from each subject a signed and informed agreement that they agreed their subjective opinions would be used for non-commercial research, making it equipped with such legal and ethical characteristics.

For the recognition task, we divide the whole subjective experiment into two stages considering the workload and avoiding visual fatigue. In the first stage, participants are asked to describe the provided image contents and determine whether it is an OBI image, thus completing the *What* and *Yes-or-No* queries, as depicted in Fig. 5. In the second stage, participants are told to delineate the positions of individual oracle bone characters as they perceive them. We adopt the commonly used toolbox "LabelImg" (Tzutalin, 2015) for annotation. The resulting number of bounding boxes and the corresponding coordinates are used for evaluating the performance under *How* and *Where* queries. As for the deciphering task, participants are told to write down their understanding of each oracle bone character, which is then taken to compare with the *golden* description for alignment evaluation. The user interface is the same as the recognition task (Fig. 5). Overall, it took over a week to complete the whole subjective experiment, where each participant contributed an average of 14.8 hours to attend.

## B  MORE INFORMATION ON BENCHMARK DATASETS

### B.1  THE PROPOSED O2BR DATASET

The objective of recognizing oracle bone inscription lies in accurately locating each single character on original oracle bones, rubbings, or fragments, thus assisting downstream classification or deciphering tasks (Xing et al., 2019). In the past decade, a variety of datasets (Guo et al., 2015; Huang et al., 2019; Han et al., 2020; AYNU, 2020; Zhang et al., 2021a; Yue et al., 2022; Wang et al., 2022) have been established for oracle bone character recognition. Among them, Oracle-20k (Guo et al., 2015) was proposed under a controllable lab environment with a total of 20,039 synthesized OBI images in 291 categories. Later, Oracle-50k (Han et al., 2020) was released with more diverse character classes and images. In (Zhang et al., 2021a), a large-scale dataset with character-level annotations, OracleBone-8000, was collected, which contains 7,824 OBI rubbing images scanned from (Guo, 1982) and 128,770 annotated character crops. Similarly, YinQiWenYuan-detection (AYNU, 2020) dataset contributed 9,306 annotated OBI rubbing images with character-level bounding boxes. However, the image contents of the above OBI recognition datasets are all handprinted or rubbing form, rather than original oracle bone. More importantly, the majority of datasets either lack the location information for individual oracle characters or do not release the data publicly. To address these problems, we propose the Original Oracle Bone Recognition (**O2BR**) dataset, which consists of 800 carefully curated original oracle bone images and 4,211 bounding boxes.

### B.1.1 SOURCE CONTENT COLLECTION

To curate the O2BR dataset, we began with the initial filtering of available websites using hashtags commonly associated with oracle bone image, such as `#oracle bone archaeology`, `#shell and bone script`, and `#ancient character`. This process allowed us to obtain a set of candidate websites on oracle collection supported by civic organizations, museums, and research institutes around the world. Considering the type, quantity, and usage requirements of the images, we choose to download original oracle bone images from the website[1] of open museum led by the Institute of History & Philology, Academia Sinica. Out of the potentially accessible retrieved results, around thirty thousand contained downloadable and visible OBI photos as of July 2024. We utilized a Python-based crawler to download a total of 21,453 raw images. Note that all images downloaded from the website are released under an appropriate Creative Commons (CC) license that allows further editing and redistribution. After that, two graduate students with oracle research backgrounds were involved in removing small fragments that do not contain complete oracle bone characters. Finally, 800 original oracle bone images are selected as the annotation source. Fig. 6 presents several samples of this dataset. Tab. 7 provides detailed statistics.

### B.1.2 ANNOTATION PROCESS

Some studies (Li et al., 2020; Diao et al., 2023) suggested the use of automatic object detection methods to save manual annotation effort. However, considering the particularity of the image content in this dataset and the lack of relevant trainable datasets, we adopted the commonly used toolbox "LabelImg" (Tzutalin, 2015) to conduct the character-level annotation. Fig. 7(a) shows the annotation interface. Two OBI domain experts participated in this process. Besides, to avoid duplication of evaluation tasks in OBI-Bench, we merely required the annotator to frame the location of a single oracle bone character. Afterward, a senior OBI domain expert was invited to proofread the annotations. If there is a disagreement between two experts, the senior expert will re-annotate the image and determine the final annotation, which helps to guarantee the quality of our proposed dataset.

Table 7: Statistics of the proposed O2BR dataset.

| Attribute | Value (in pixels) |
|---|---|
| Minimum Width | 167 |
| Maximum Width | 2167 |
| Minimum Height | 168 |
| Maximum Height | 2664 |
| Average Width | 1529 |
| Average Height | 1192 |
| Avg. number of bounding boxes/image $\approx 5.264$ | |

### B.2 THE PROPOSED OBI-REJOIN DATASET

In the early days, the most important reference in OBI domain is the "Collections of oracle bone inscriptions" book (Guo, 1982), which contains nearly 2,000 groups of OBI rejoining results. Other representative books such as "Catalog of Oracle Bone Rejoinings" (Tsai, 1999), "Zui Gu: Research on Oracle Bone Rejoinings" (Lin, 2008), "The Fifth Collection of Oracle Bone Rejoinings" (Huang, 2019), and "Compilation of Oracle Bone Inscriptions: Reconstruction and Interpretation" (Zhang, 2020) involve the achievements of over 4,000 groups of OBI rejoining results in the past 30 years. However, there is no publicly available rejoining dataset for any type of OBIs in digital image form. Recently, researchers collected a real-world dataset for rejoining Oracle Bone fragments (Zhang et al., 2022), which consists of 998 oracle bone rubbing images with manually outlined top and bottom marginal areas. Unfortunately, its proprietary nature limits its use by the community. This motivates us to build the **OBI-rejoin**, an OBI dataset comprising 200 complete OBI pieces with 483 rejoinable fragments, shown in Fig. 8. Tab. 8 lists the statistical information of OBI-rejoin dataset.

Table 8: Statistics of the proposed OBI-rejoin dataset.

| Attribute | Value (in pixels) | |
|---|---|---|
| | map | fragments |
| Minimum Width | 64 | 46 |
| Maximum Width | 1268 | 909 |
| Minimum Height | 107 | 29 |
| Maximum Height | 2913 | 1273 |
| Average Width | 370 | 265 |
| Average Height | 465 | 313 |
| Avg. number of fragments/map = 2.415 | | |

### B.2.1 SOURCE CONTENT COLLECTION

We used a Python-based crawler to scrape metadata from the official website[2] of the Pre-Qin Research Office at the Institute of History, Chinese Academy of Social Sciences. This process covered 217

---

[1] https://openmuseum.tw/objects
[2] https://www.xianqin.org/blog/archives/category/jgw_study/jgw_zhuihe

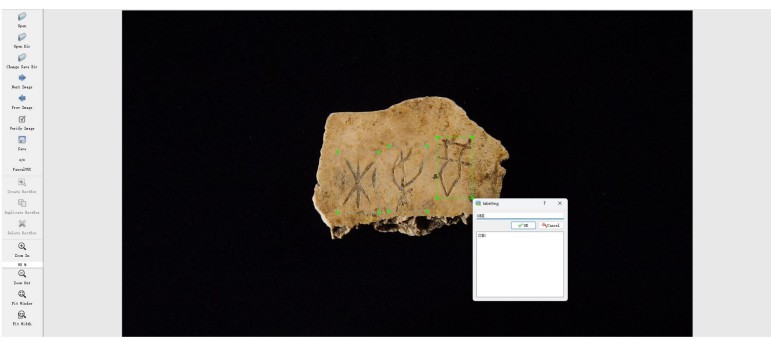

(a) Image annotation interface for the O2BR dataset

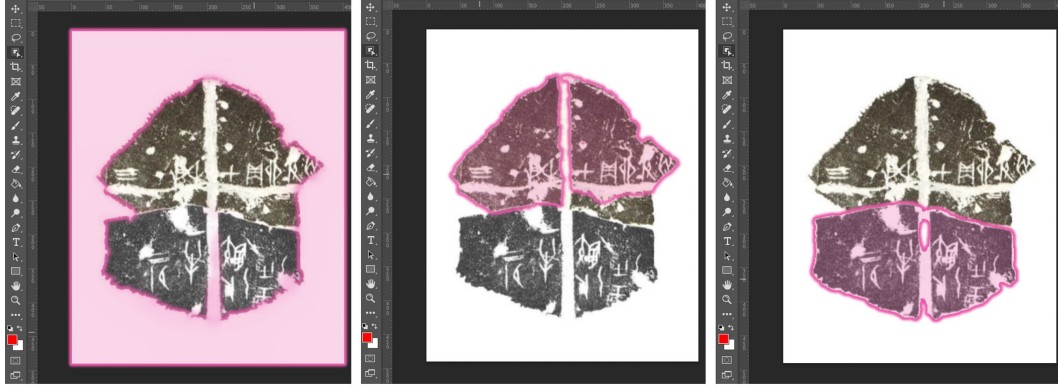

(b) Image segmentation interface for the OBI-rejoin dataset

Figure 7: The illustration of the annotation interfaces for the **O2BR** dataset (coordinate) on recognition task, and the **OBI-rejoin** dataset (adjacency) on rejoining task.

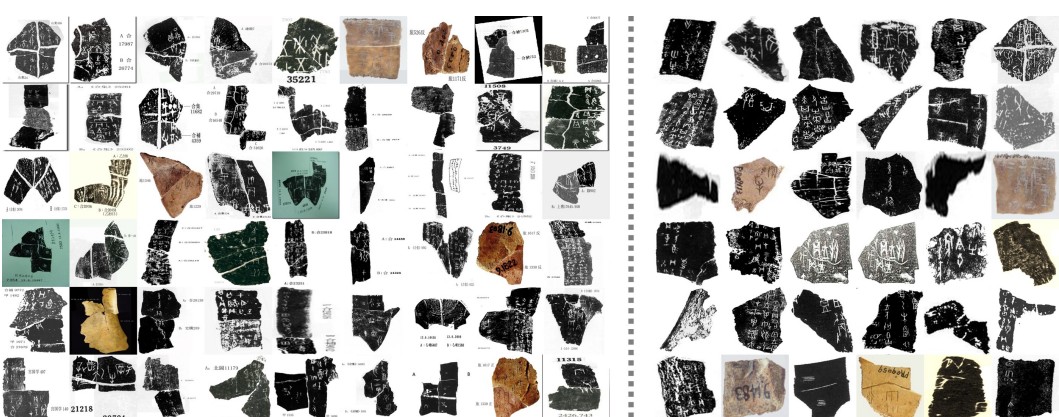

Figure 8: Sample images of the original OBI rejoining results (left) as well as the separated fragments (right) in the OBI-rejoin dataset.

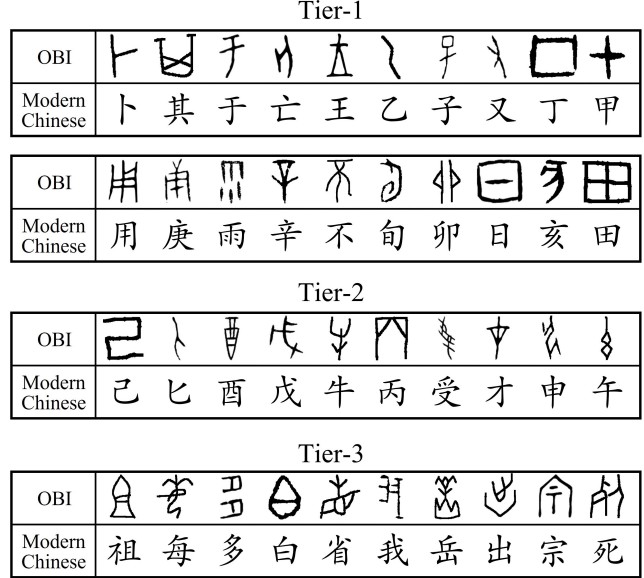

Figure 9: Visualization of 40 oracle bone characters with different frequency levels.

pages and encompassed the time frame from December 2005 to July 2024, resulting in 4,249 OBI images[3]. To ensure data quality, we conducted an expert examination that manually excluded hand-drawn contour images as well as extraneous contents, retaining only inked rubbings and original oracle bone forms as the image sources of the OBI-rejoin dataset. As a result, a total of 200 OBI images covering two forms of OBI (*i.e.*, original and inked) with different character styles and resolutions are selected.

### B.2.2 ANNOTATION PROCESS

Here, we refer to the OBI rejoining results as "map" to reflect its special role in the construction process of OBI-rejoin dataset. Three domain experts in OBI research and one senior OBI scholar were involved in the annotation process of OBI-rejoin. Each annotator was equipped with a PC with commercial photo editing software. Fig. 7(b) illustrates the annotation interface. To precisely reproduce the authentic conditions of oracle bone fractures, the annotator manually tears "maps" on appropriate borders utilizing their domain expertise to induce central or peripheral fractures. A double-check procedure was applied after the annotation to ensure quality and rationality. Overall, it took each annotator approximately 3 hours to finish annotation, and the whole construction process took over a week to complete. A lot of effort and discussion were required in order to manually retrieve suitable rejoining results from the crawled metadata. To the best of our knowledge, **OBI-rejoin** is the *first* open-source oracle bone inscription rejoining dataset. We hope that OBI-rejoin can facilitate future development of advancing OBI rejoining algorithms and serve as a benchmark dataset to evaluate the performance of rejoining methods.

### B.3 COLLECTING GOLDEN DESCRIPTIONS FOR DECIPHERING TASK

The existing OBI deciphering works merely decipher oracle bone characters by linking them to modern Chinese characters (Chang et al., 2022; Guan et al., 2024b), which does not constitute a true decipherment. With the historical and cultural changes, many oracle bone characters can no longer find correspondence with modern Chinese characters. Thus, we turn to the most traditional dictionary form and extract *golden* descriptions from an authoritative Chinese character database (CUHK, 2014).

---

[3]We extend our deepest gratitude to the frontline researchers and scholars involved in the meticulous collation and proofreading of the oracle bone inscriptions.

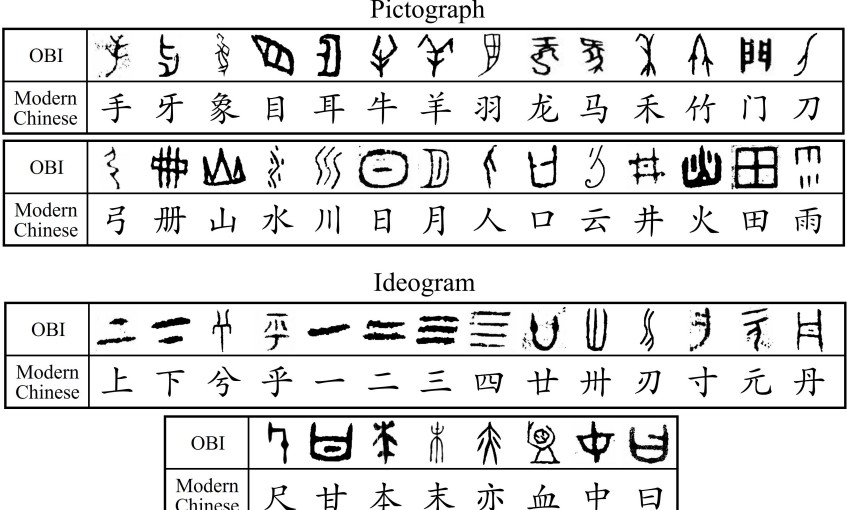

Figure 10: Visualization of 28 and 22 oracle bone characters in pictograph and ideogram formation, respectively.

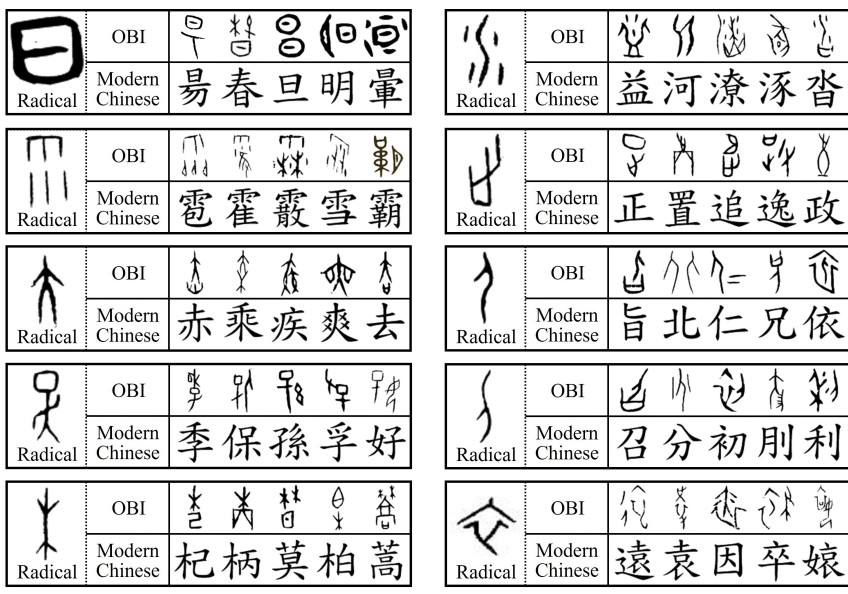

Figure 11: Visualization of 10 different radicals with their structural variants as well as the corresponding modern Chinese characters.

Table 9: A brief view of 17 open-source LMMs evaluated in the **OBI-Bench** in *chronological* order.

| *Date of Release* Model Names | Vision Architectures (V) | | V→L | Language Architectures (L) | |
| --- | --- | --- | --- | --- | --- |
| | Backbone | #Size | Alignment | Backbone | Type |
| [24.08]xGen-MM-instruct-interleave (Xue et al., 2024) | SigLIP-400M | 384 | Perceiver resampler | Phi-3 mini | *decoder-only* |
| [24.08]mPLUG-Owl3-7B (Ye et al., 2024) | SigLIP-400M | 384 | MLP projector | Qwen2-7B | *decoder-only* |
| [24.08]MiniCPM-V 2.6-8B (Yao et al., 2024) | SigLIP-400M | 224 | Perceiver resampler | Qwen2-7B | *decoder-only* |
| [24.08]Moondream2-1.6B (Moondream.ai, 2024) | SigLIP-400M | 378 | MLP projector | Phi-1.5 | *decoder-only* |
| [24.07]InternVL2-Llama3-76B (Chen et al., 2024a) | InternViT-6B | 448 | MLP projector | Hermes-2-Theta-Llama-3-70B | *decoder-only* |
| [24.07]InternVL2-40B (Chen et al., 2024a) | InternViT-6B | 448 | MLP projector | Nous-Hermes-2-Yi-34B | *decoder-only* |
| [24.07]InternVL2-8B (Chen et al., 2024a) | InternViT-300M | 448 | MLP projector | InternLM2.5-7B | *decoder-only* |
| [24.06]GLM-4V-9B (GLM et al., 2024) | EVA2-CLIP-E/14+ | 224 | MLP projector | GLM-4-9B | *decoder-only* |
| [24.05]CogVLM2-LLaMA3-Chat-19B (Wang et al., 2023) | EVA2-CLIP-E | 224 | MLP projector | Llama-3-8B-Instruct | *decoder-only* |
| [24.05]LLaVA-NeXT-72B (Li et al., 2024a) | CLIP-ViT-L/14 | 336 | MLP projector | Qwen1.5-72B | *decoder-only* |
| [24.05]LLaVA-NeXT-8B (Li et al., 2024a) | CLIP-ViT-L/14 | 336 | MLP projector | Llama-3-8B-Instruct | *decoder-only* |
| [24.05]IDEFICS-2-8B Laurençon et al. (2024) | SigLip-400M | 384 | MLP projector | Mistral-7B | *decoder-only* |
| [24.03]DeepSeek-VL-7B (Lu et al., 2024) | SAM-B&SigLIP-L | 1024&384 | MLP projector | DeepSeek-LLM-7B | *decoder-only* |
| [24.01]InternLM-XComposer2-VL-7B (Dong et al., 2024) | CLIP-ViT-L/14 | 336 | Partial LoRA | InternLM2-7B | *decoder-only* |
| [23.10]LLaVA-v1.5-13B (Liu et al., 2024a) | CLIP-ViT-L/14 | 336 | MLP projector | Vicuna-v1.5-13B | *decoder-only* |
| [23.10]LLaVA-v1.5-7B (Liu et al., 2024a) | CLIP-ViT-L/14 | 336 | MLP projector | Vicuna-v1.5-7B | *decoder-only* |
| [23.08]Qwen-VL (Bai et al., 2023) | CLIP-ViT-G/14 | 448 | Cross-attention | Qwen-7B | *decoder-only* |

# C ADDITIONAL EXPERIMENTAL RESULTS

## C.1 BENCHMARK CANDIDATES

To ensure the comprehensiveness and timeliness of the results, we select **23** up-to-date and prevailing LMMs for evaluation. The **proprietary LMMs** include Gemini 1.5 Pro (Reid et al., 2024), Gemini 1.5 Flash (Reid et al., 2024), GPT-4v (Achiam et al., 2023), GPT-4o (OpenAI, 2024a), Qwen-VL-Max (Bai et al., 2023), and GLM-4v (GLM et al., 2024). The **open-source LMMs** include xGen-MM-instruct-interleave (Xue et al., 2024), mPLUG-Owl3-7B (Ye et al., 2024), MiniCPM-V 2.6-8B (Yao et al., 2024), Moondream2-1.6B (Moondream.ai, 2024), InternVL2-Llama3-76B (Chen et al., 2024a), InternVL2-40B (Chen et al., 2024a), InternVL2-8B (Chen et al., 2024a), GLM-4V-9B (GLM et al., 2024), CogVLM2-LLaMA3-Chat-19B (Wang et al., 2023), LLaVA-NeXT-72B (Li et al., 2024a), LLaVA-NeXT-8B (Li et al., 2024a), IDEFICS-2-8B Laurençon et al. (2024), DeepSeek-VL-7B (Lu et al., 2024), InternLM-XComposer2-VL-7B (Dong et al., 2024), LLaVA-v1.5-13B (Liu et al., 2024a), LLaVA-v1.5-7B (Liu et al., 2024a), and Qwen-VL (Bai et al., 2023). Tab. 9 summarizes and compares the vision and language architectures of open-source LMMs.

Additionally, considering the cultural uniqueness of OBI, and to ensure the representativeness of human performance on the OBI-Bench, we recruited five Chinese college students with no background in the ancient Chinese language to reflect public cognition. Initially, we instructed all participants to have a clear and consistent understanding of all question-answer patterns by familiarizing themselves with the tasks through exposure to similar cases. Note that we conduct user studies in the same order of tasks as we test LMMs to avoid interference between tasks. Given the inherent variability of LMMs, identical prompts can yield responses. To address the impact of such situations on our evaluation, we implemented a 5-round average strategy.

## C.2 ADDITIONAL DETAILS OF EVALUATION ON RECOGNITION

### C.2.1 EVALUATION CRITERIA

For *What* question, we calculate the cosine similarity between the embeddings of output answers and the *golden* descriptions. Specifically, we use the `Sentence-Transformers` python library and the `all-MiniLM-L6-v2`[4] model, which maps sentences to a 384-dimensional dense vector space.

For *How* question, we use the relative counting error metric to measure the performance of different LMMs:

$$\text{MRE} = \frac{1}{N} \sum_{i=1}^{N} \frac{\left| C_i^{truth} - C_i^{pred} \right|}{C_i^{truth}} \tag{1}$$

where $N$ is the number of evaluated OBI images. $C_i^{truth}$ and $C_i^{pred}$ denote the actual number of oracle bone characters in the image and the predicted number of oracle bone characters respectively.

---

[4]https://huggingface.co/sentence-transformers/all-MiniLM-L6-v2

Table 10: Impact of the judgment bias. Answer accuracy of LMMs on *Yes-or-No* questions.

| LMM | Yes-or-No |
|---|---|
| GEMINI 1.5 PRO | 96.15% |
| GEMINI 1.5 FLASH | 93.37% |
| GPT-4V | 95.13% |
| GPT-4O | 98.68% |
| QWEN-VL-MAX | 92.26% |
| GLM-4V | 86.68% |
| xGen-MM (*Instruct-interleave-4B*) | 83.55% |
| mPLUG-Owl3 (*Qwen2-7B*) | 87.68% |
| MiniCPM-V 2.6 (*Qwen2-7B*) | 82.21% |
| InternVL2-Llama3-76B (*Llama3-70B*) | 93.59% |
| InternVL2-40B (*Nous-Hermes2-Yi-34B*) | 92.24% |
| InternVL2-8B (*InternLM2.5-7B*) | 86.55% |
| GLM-4V-9B (*GLM-4-9B*) | 81.62% |
| CogVLM2-Llama3-19B (*Llama3-8B*) | 88.68% |
| LLaVA-NeXT (*Qwen1.5-72B*) | 94.77% |
| LLaVA-NeXT (*Llama3-8B*) | 85.50% |
| IDEFICS-2-8B (*Mistral-7B*) | 87.55% |
| DeepSeek-VL (*DeepSeek-LLM-7B*) | 84.37% |
| InternLM-XComposer2-VL (*InternLM2-7B*) | 86.88% |
| LLaVA-v1.5 (*Vicuna-v1.5-13B*) | 79.93% |
| LLaVA-v1.5 (*Vicuna-v1.5-7B*) | 75.58% |
| Qwen-VL (*Qwen-7B*) | 84.57% |

### C.2.2 EXTENDED RESULTS ON YES-OR-NO QUESTION

Considering that the test set used in the recognition task is composed solely of authentic oracle bone rubbings or fragments, it may be affected by the response preferences of LMMs in Yes-or-No questions. In this section, we take a deeper analysis of the Yes-or-No judgment ability of LMMs, and whether these models can get similar accuracy on questions that should be answered with 'Yes', as those should be replied to as 'No'. Specifically, we re-curate a content-mixed test set, which consists of 1,000 OBI images and other images containing ancient texts (such as the inscription on the bronzes, characters engraved on bamboo slips, and characters written on silk). As a result, the proportion of OBI images and non-OBI images is 1:1. As reported in Tab. 10, the performance of GLM-4V and GPM-4V-9B improves greatly compared to the results in the O2BR and YinQiWenYuan$_{detection}$ datasets, while the other LMMs perform at a similar level.

### C.3 ADDITIONAL DETAILS OF EVALUATION ON CLASSIFICATION

To explore the effect of OBI image quality on classification accuracy, we further select two classical image quality assessment (IQA) metrics, namely NIMA (Talebi & Milanfar, 2018) and CNNIQA (Kang et al., 2014), to measure the average image quality within each test sets. As listed in Tab. 11, the numerical results of both metrics show that the image quality of HWOBC and Oracle-50k datasets is better than that of OBI125. A similar conclusion to the main paper can be drawn that enhancing the image quality or applying denoising operation can improve the OBI classification accuracy of LMMs (Chen et al., 2025).

### C.4 ADDITIONAL DETAILS OF EVALUATION ON DECIPHERING

#### C.4.1 EFFECT OF PRE-PROMPT

Existing studies (Chen et al., 2023; Zhou et al., 2023) indicate that providing specific instructions, contextual information, or role settings allows the model to understand the user's needs and expectations and thus provide a more accurate and relevant response. We thereby employ two schemes (*i.e.*, role assignment and case instruction) to investigate the effect of pre-prompt.

**Prompt Template for Role Assignment:**

Table 11: Image quality comparison between two handprinted OBI datasets and one cropped from real rubbings.

| Metric | HWOBC | Oracle-50k | OBI125 |
|---|---|---|---|
| NIMA↑ | 5.3003 | 3.9973 | 3.6710 |
| CNNIQA↑ | 0.7165 | 0.6229 | 0.5612 |

Table 12: Effect of pre-prompt on the **OBI deciphering** task. We select Gemini 1.5 Pro, GPT4o, Qwen-VL-Max, and GLM-4V for evaluation.

| Datasets | HUST-OBS | | | EVOBC | | OBI Component 20 | Average |
|---|---|---|---|---|---|---|---|
| LMM (*variant*) | *Tier-1* | *Tier-2* | *Tier-3* | *pictograph* | *ideogram* | *Radical* | |
| GEMINI 1.5 PRO | 0.3766 | 0.3834 | 0.3589 | 0.4226 | 0.3696 | **0.3750** | 0.3810 |
| GEMINI 1.5 PRO+Role | 0.3824 | 0.3863 | 0.3603 | 0.4347 | **0.3819** | 0.3738 | 0.3866 |
| GEMINI 1.5 PRO+Role+Case | **0.3837** | **0.3867** | **0.3626** | **0.4386** | 0.3779 | 0.3722 | **0.3870** |
| GPT-4O | 0.3891 | 0.3510 | 0.3660 | 0.4535 | **0.3893** | 0.3767 | 0.3876 |
| GPT-4O+Role | **0.3925** | 0.3525 | 0.3557 | **0.4585** | 0.3817 | 0.3729 | 0.3856 |
| GPT-4O+Role+Case | 0.3922 | **0.3586** | **0.3688** | 0.4545 | 0.3829 | **0.3783** | **0.3892** |
| QWEN-VL-MAX | 0.2345 | 0.2273 | 0.2322 | 0.2565 | 0.2533 | 0.2785 | 0.2471 |
| QWEN-VL-MAX+Role | 0.2447 | 0.2535 | 0.2533 | 0.2585 | 0.2401 | 0.2796 | 0.2549 |
| QWEN-VL-MAX+Role+Case | **0.2518** | **0.2778** | **0.2617** | **0.2830** | **0.2457** | **0.2883** | **0.2681** |
| GLM-4V | 0.2180 | 0.1638 | 0.2122 | 0.2353 | 0.2451 | 0.1683 | 0.2071 |
| GLM-4V+Role | 0.1977 | **0.1984** | 0.1989 | 0.2554 | 0.2448 | 0.1689 | 0.2107 |
| GLM-4V+Role+Case | **0.2187** | 0.1968 | **0.2137** | **0.2585** | **0.2487** | **0.1710** | **0.2179** |

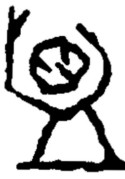 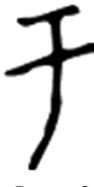 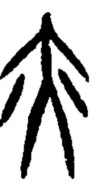

<Image1>   <Image2>   <Image3>

Figure 12: Example of three oracle bone character images that are taken as case instructions along with their *golden* descriptions.

*#System: You are a senior oracle bone researcher who excels in interpreting ancient texts. Please help me to predict the meaning of the undeciphered oracle bone characters reasonably and accurately.*

*#User: What is the meaning of this oracle bone character? <image>*

**Prompt Template for Case Instruction:** We further randomly select three cases (OBI images with correct descriptions) as case instructions (Fig. 12) after role assignment.

*#System: Here are three examples of deciphered oracle bone characters.*

*#System: (1) <image1> The oracle bone character in this image means the blood, which also represents the blood of animals offered to the gods in sacrifices.*

*#System: (2) <image2> The oracle bone character in this image refers to the pattern and texture of the bamboo tubes merged together on a musical instrument.*

*#System: (3) <image3> The oracle bone character in this image means the armpit, with the small dots indicating the armpits. It is like a person sweating in his armpits.*

*#System: Please refer to the case form given above for the following answers.*

*#User: What is the meaning of this oracle bone character? <image>*

In Tab. 12, we compare the deciphering performance of LMMs with or without pre-prompt operations. It can be observed that adding role assignment and case instruction slightly improves the accuracy and reasonableness of decipherment results for Gemini 1.5 Pro, GPT-4o, Qwen-VL-Max, and GLM-4V by an average of 2.02%, 0.40%, 8.50%, and 5.21% respectively. This demonstrates the effectiveness and necessity of prompt engineering in refining the answers of LMMs and avoiding meaningless outputs.

### C.4.2 DECIPHERING RESULTS ON UNKNOWN OBIS

We provide the deciphering results of LMMs for previously undeciphered five oracle bone characters from the HUST-OBS dataset (Wang et al., 2024a), whose interpretation is central to the core of

Table 13: Taxonomy of LMMs categorized by the affiliation of the first author.

| Model | Affiliation of the First Author | Country |
|---|---|---|
| **Proprietary LMMs:** | | |
| GEMINI 1.5 PRO | Google Inc. | USA |
| GEMINI 1.5 FLASH | Google Inc. | USA |
| GPT-4V | OpenAI Inc. | USA |
| GPT-4O | OpenAI Inc. | USA |
| QWEN-VL-MAX | Alibaba Group | China |
| GLM-4V | Zhipu AI Inc. | China |
| **Open-source LMMs:** | | |
| xGen-MM (*Instruct-interleave-4B*) | Salesforce AI Research | USA |
| mPLUG-Owl3 (*Qwen2-7B*) | Alibaba Group | China |
| MiniCPM-V 2.6 (*Qwen2-7B*) | ModelBest Inc. | China |
| Moondream2 (*ver. 0728*) | M87 Labs | USA |
| InternVL2-Llama3-76B (*Llama3-70B*) | Shanghai AI Lab | China |
| InternVL2-40B (*Nous-Hermes2-Yi-34B*) | Shanghai AI Lab | China |
| InternVL2-8B (*InternLM2.5-7B*) | Shanghai AI Lab | China |
| GLM-4V-9B (*GLM-4-9B*) | Zhipu AI Inc. | China |
| CogVLM2-Llama3-19B (*Llama3-8B*) | Tsinghua University | China |
| LLaVA-NeXT (*Qwen1.5-72B*) | University of Wisconsin-Madison | USA |
| LLaVA-NeXT (*Llama3-8B*) | University of Wisconsin-Madison | USA |
| IDEFICS-2-8B (*Mistral-7B*) | Hugging Face Inc. | USA |
| DeepSeek-VL (*DeepSeek-LLM-7B*) | DeepSeek-AI Inc. | China |
| InternLM-XComposer2-VL (*InternLM2-7B*) | Shanghai AI Laboratory | China |
| LLaVA-v1.5 (*Vicuna-v1.5-13B*) | University of Wisconsin–Madison | USA |
| LLaVA-v1.5 (*Vicuna-v1.5-7B*) | University of Wisconsin–Madison | USA |
| Qwen-VL (*Qwen-7B*) | Alibaba | China |

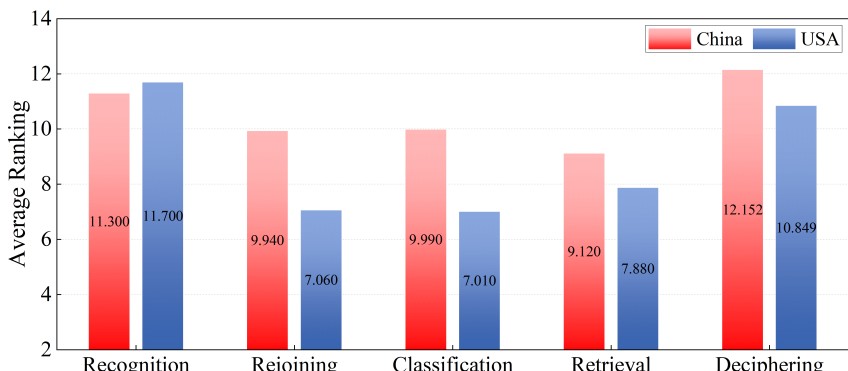

Figure 13: Comparison of the average ranking of LMMs under region-based taxonomy.

Pre-Qin cultural and historical research, in Fig. 14, Fig. 15, Fig. 16, Fig. 17, and Fig. 18. Different from existing decipherment studies (Zhang et al., 2021b; Guan et al., 2024b) that decipher oracle bone characters as characters in subsequent writing systems or pseudo-modern Chinese characters, we directly leverage the multi-modal capabilities of LMMs to decipher them as a description in English. Since the correctness of an oracle decipherment result is usually determined by peer review, we plan to make the code used in OBI-Bench and a comprehensive collection encompassing more decipherment outcomes publicly available. We hope this contribution will assist OBI researchers in advancing the study of ancient languages.

## C.5 IMPACT OF POTENTIAL CULTURE BIAS

We further investigate the impact of different cultural backgrounds on OBI processing performance. Tab. 13 classifies LMMs by the affiliation of the first author. Among all evaluated LMMs, there are 12 LMMs from China and 11 LMMs from the USA. To ensure the fairness of the results, we exclude the earliest LMM (Qwen-VL) to make the number of models the same. Given the differences in evaluation criteria and scales of different tasks, we report the average ranking of LMMs according to

the region-based taxonomy, shown in Fig. 13. We find that cultural differences did not particularly affect the overall performance of the models. Besides, the LMMs originating from China still have some distance behind those from the United States.

## D    LIMITATIONS

Although our study provides a comprehensive evaluation of LMMs in OBI tasks, there are several potential limitations acknowledged below. First, the images in our test sets are limited in scope with regard to size, content types for each task, and resolutions, which constrains the applicability of the OBI-Bench. We merely sample partial images from YinQiWenYuan$_{detection}$, HWOBC, Oracle-50k, OBI125, OBC306, OBI-IJDH, EVOBC, OBI Component 20, and HUST-OBS datasets as test sets. Our self-collected O2BR and OBI-rejoin datasets contain 800 and 483 images. Besides, in the deciphering task, we selected only 140 oracle bone characters for evaluation, which is considerably fewer than the number of deciphered oracle bone characters currently known. Another limitation of our work is that the query-answer forms used in our benchmark are relatively simple. There is no extra information that gives the LMMs clues to the intent of the image. The lack of such context hinders performance in cases such as OBI deciphering where context (role assignment and case instruction), has been shown to aid in image interpretation (App. C.4.1). Furthermore, coarse-grained perception perspectives such as *What* query in the recognition task and deciphering task involve subjectivity and should be based on general consensus while defining golden descriptions.

These limitations highlight the need for related future research. We encourage the community to view our work as a starting point and extend the evaluations and analysis to further uncover potential capabilities of LMMs not only in the OBI domain but also in other ancient character processing domains and design possible task-oriented practical strategies accordingly.

## E    FUTURE POSSIBILITIES

Apart from the comments for possible next steps of research related to LMMs and OBI that have already been given, this section is devoted to the extension for some of them and then more topics with good potential based upon our understanding and rethinking for the field.

New possibilities of LMM in aiding OBI processing advancement can be tried in a number of major directions:

- **Focus on Robust Preprocessing Pipelines:** Developing preprocessing techniques that can enhance low-quality or noisy OBI data, such as denoising, fragment reconstruction, or super-resolution methods, may significantly improve LMM's performance on these tasks.
- **Domain-Specific Fine-Tuning with Interdisciplinary Collaboration:** Fine-tuning models on domain-specific datasets can help bridge the gap in domain knowledge. Integrating visual and textual modalities can enrich the understanding of oracle bone inscription, thus enhancing the overall interpretative capability. Bridging the gap between computer science and humanities disciplines is crucial for effective application of LMMs in OBI studies. Collaborating with historians and linguists to create multi-modal annotated OBI datasets may benefit this effort.
- **Interactive Question Answering Systems:** Future applications could include interactive question-answering systems where users can inquire about oracle bone inscription meanings and historical contexts in natural language, leveraging large multi-modal models for intuitive and accessible interactions.

It is hoped that these can provide actionable ideas for researchers and would trigger further discussion, and more importantly, new exploration in this field.

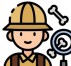 **OBI researcher**:
*Prompt (Prefix):*
You are a senior oracle bone researcher who excels in interpreting ancient texts. Please help me to predict the meaning of the undeciphered oracle bone characters reasonably and accurately.
*Prompt (Main):*
What is the meaning of this oracle bone character? *<image>* 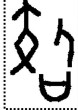

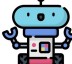 **GPT-4o**:
*Structural Elements:*
The character has a top part that resembles a roof-like structure or a person standing, a central part that looks like an enclosed or diamond shape, and a lower section resembling a container or mouth shape.
*Possible Interpretations:*
Enclosure or Containment: The diamond-like structure could symbolize something contained or enclosed, often seen in characters that imply boundaries, houses, or objects held within.
Communication or Exchange: The mouth-like shape at the bottom might suggest communication, a mouth, or something related to speech or verbal exchange.
Human Figure: The top part, somewhat resembling a person, could indicate involvement of a human or anthropomorphic representation.
*Potential Meanings:*
This combination could imply a concept related to ritual activities, such as "speaking within boundaries" (potentially linked to offerings or ceremonial actions).
It might also indicate a title or a specific role within a ritual context, possibly a "messenger" or "interpreter" who has a function within a defined space.
Given the structure, the character might symbolize a ceremonial or communicative role within a ritual.

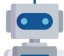 **Gemini 1.5 Pro**:
The character appears to be a combination of three components:
Top: A vertical line with two short horizontal strokes branching out at the top. This element often represents a person or human figure.
Middle: A criss-cross or knotted pattern. This element might signify binding, tying, or connecting.
Bottom: A U-shape, which could depict a container, vessel, or receptacle.
*Possible Interpretations:*
Considering the combination of these elements, here are a few plausible interpretations:
Offering or Sacrifice: The human figure above the knotted pattern and container could represent someone presenting an offering or sacrifice in a vessel.
Binding or Confinement: The knotted pattern could suggest binding or restricting something within the container.
Storage or Preservation: The container with the figure above might indicate storing or preserving something valuable or important.

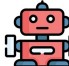 **GLM-4V**:
The character depicted here, if it's an oracle bone inscription, appears to be composed of two elements:
The top part looks like "⿺," which can mean "three" or be part of other words with different meanings depending on context. It also resembles the character for horse (馬), but without the four legs, which might suggest a stylistic variation or a different meaning here. The bottom part is "口," which means "mouth."
Combining these elements doesn't form a known word in classical Chinese, so this character may be a unique combination or a variant that has lost its original meaning over time. Inscriptions of this nature often require additional context from similar inscriptions or historical records to accurately interpret.

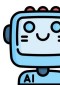 **Qwen-VL-Max**:
The character you provided appears to be a combination of several elements, which might represent a specific concept or object. The character on the left seems to depict a stylized form of a plant or a tree, which is common in oracle bone script. The character on the right appears to be a simplified representation of a hand or a tool. Given these elements, the combined character might represent a concept related to agriculture or farming, such as "plowing" or "cultivating." However, without more context or additional characters, it's difficult to provide a definitive interpretation. Oracle bone characters often require a broader context to understand their full meaning

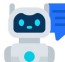 **MiniCPM-V 2.6**:
The oracle bone character shown in the image is a representation of the concept "mutual trust" or "reciprocal trust." This interpretation comes from the common understanding of such characters, which often convey abstract ideas and concepts. The visual elements suggest an agreement or mutual commitment between two parties, as indicated by the crossed lines that resemble arms or hands reaching out to each other.

 **Moondream2**:
The oracle bone character in the image is a symbol used in ancient Chinese culture to represent the concept of knowledge and guidance. It is often associated with the idea of seeking wisdom and insight from the past or the future. The character is typically depicted as a stylized representation of a bone, which is a symbol of strength and durability. In the context of the image, the oracle bone character could be interpreted as a symbol of the importance of learning from history and seeking guidance from the past to make informed decisions in the present or to anticipate the future.

**LLaVA-NeXT-8B**:
The image you've provided appears to be an oracle bone character, which is an ancient Chinese character carved on animal bones or turtle shells. Oracle bone characters are a form of divination that was used in ancient China to predict the future and to communicate with the gods. The specific meaning of the character would depend on the context in which it was used, as well as the interpretation of the diviner. Oracle bone characters are no longer used for divination in modern times, but they are still studied by scholars for their historical and cultural significance.

Figure 14: A qualitative comparison for the deciphered results of 7 representative LMMs on a genuine undeciphered oracle bone character (I).

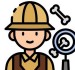 **OBI researcher**:
*Prompt (Prefix):*
You are a senior oracle bone researcher who excels in interpreting ancient texts. Please help me to predict the 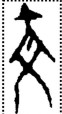 meaning of the undeciphered oracle bone characters reasonably and accurately.
*Prompt (Main):*
What is the meaning of this oracle bone character? *<image>*

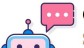 **GPT-4o**:
*Structural Elements:*
The character appears to depict a person or human figure, often seen in Oracle Bone script to represent actions, roles, or identities. The presence of what looks like a hat or a specific head shape might indicate a particular person or role, such as a leader, shaman, or figure of authority.
*Comparative Analysis:*
Similar human-like forms in Oracle Bone inscriptions are often associated with roles in rituals, ancestral worship, or specific societal functions like diviners or warriors. If the head shape suggests a crown or distinctive mark, it could represent a significant person, such as a king or noble.
*Contextual Implications:*
If this character is found alongside characters related to ritual, divination, or sacrifice, it could denote a person involved in the ritual, possibly a priest or king. The overall posture and design suggest a figure of importance, likely standing or commanding, which can indicate leadership or an authoritative role.
*Symbolic Meaning:*
It is possible that this character signifies "person" (人), "chief" (长), or a ritual participant, depending on the context within which it is found. The specific headdress or symbol on top might further differentiate this figure from a generic "person" symbol, emphasizing a specialized role or identity.
*Reasonable Prediction:*
Based on the visual analysis and typical representations of human figures in Oracle Bone script, this character likely represents a person of significance, possibly a leader, ritual participant, or an important societal figure. If more context from surrounding inscriptions is available, it would help refine the exact role or meaning.

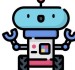 **Gemini 1.5 Pro**:
*Potential Pictographic Interpretations:*
Human Figure: The most apparent interpretation is that it represents a stylized human figure. The head is clearly depicted with a distinct headdress or hairstyle. The outstretched arms and legs suggest a posture of either standing, walking, or possibly dancing.
*Symbolic Possibilities:*
Leadership or Authority: The prominent headdress could signify a person of high status, a leader, or a shaman. The upright posture might further emphasize authority or a commanding presence.
Ritual or Dance: The pose could depict someone engaged in a ritualistic dance or ceremony, perhaps invoking spirits or deities.
Journey or Movement: The outstretched limbs might symbolize a journey, migration, or a transition from one state to another.

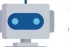 **GLM-4V**:
The character you've shown appears to be composed of two elements:
The top part resembles " 卪" (yi), which has various meanings including "to support," "to receive," or "to send."
The bottom part looks like "又" (yu), which can mean "again," "plus," or have other significations depending on the context.
However, without knowing how these elements relate to each other within the larger context or having similar characters that this character is associated with, we cannot ascertain its exact meaning.

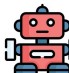 **Qwen-VL-Max**:
The oracle bone character you provided appears to be a pictographic representation. Oracle bone script is one of the earliest forms of Chinese writing, and its characters often depict objects or actions in a simplified form. The character you've shown seems to represent a person. The top part of the character resembles a head with a hat or headdress, and the lower part shows the body and legs. This type of character is likely to be a pictogram of a person, possibly indicating a specific role or status, such as a noble, a warrior, or a priest. However, without additional context or a complete sentence, it's challenging to determine the exact meaning. Oracle bone characters often need to be read in context to understand their full significance. If you have more characters or a complete inscription, it would be helpful in providing a more accurate interpretation.

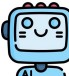 **MiniCPM-V 2.6**:
The oracle bone character shown in the image is "夫" (fū), which means husband or married man. This character is derived from ancient Chinese script and has been used for thousands of years to denote marital status, particularly in reference to a male spouse within a family context.

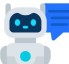 **Moondream2**:
The oracle bone character, also known as the Chinese zodiac symbol, represents the year of the year according to the Chinese zodiac. It is a symbol that combines the shape of a bone with a bird-like figure, often featuring a pointed hat. The oracle bone symbol is used to determine the year of a person's birth, and it is believed to bring good fortune and guidance. The specific meaning of the oracle bone character varies depending on the zodiac sign, but generally, it represents the year's characteristics and life lessons for that sign.

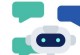 **LLaVA-NeXT-8B**:
The image you've provided appears to be an oracle bone character, which is a form of ancient Chinese writing. Oracle bone characters were used for divination and were carved onto animal bones, typically oxen, to predict the future. The specific meaning of this character would depend on the context in which it was used, as well as the interpretation of the diviner. Oracle bone characters are no longer used for divination in modern times but are still studied by scholars for their historical and linguistic significance.

Figure 15: A qualitative comparison for the deciphered results of 7 representative LMMs on a genuine undeciphered oracle bone character (II).

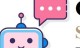 **OBI researcher**:
*Prompt (Prefix):*
You are a senior oracle bone researcher who excels in interpreting ancient texts. Please help me to predict the meaning of the undeciphered oracle bone characters reasonably and accurately.
*Prompt (Main):*
What is the meaning of this oracle bone character? *<image>* 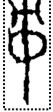

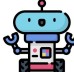 **GPT-4o**:
*Structural Elements:*
The character has a prominent vertical line running through the center, which may suggest a pole, spear, or some form of division. The upper part appears to have a branching or horn-like structure, which often symbolizes something with multiple points, branches, or limbs. The circular or rounded shape to the side of the central vertical line could represent a container, a shield, or something enclosed.
*Possible Interpretations:*
Ritual or Sacrifice Context: The vertical line and branching elements might hint at a tool or object used in ritual practices, possibly related to sacrifices or offerings.
Symbol of Authority or Protection: The vertical component could imply a staff or scepter, indicating authority, while the circular form might suggest protection or containment, such as a shield or boundary.
Animal Representation: The upper branching forms could also resemble antlers or horns, suggesting a connection to animals, possibly as a symbol or totem.
*Potential Meanings:*
This character may symbolize a concept tied to protection, authority, or ritual. It might represent a guardian figure or an object of power used in sacred practices. Alternatively, it could indicate a specific tool or item associated with rituals, especially if linked to animals or symbols of strength.

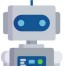 **Gemini 1.5 Pro**:
*Potential Pictographic Interpretations:*
Tree or Plant: The central vertical line with branches extending outward could represent a tree or a plant. The rounded shape at the bottom might symbolize the roots or the base of the plant.
Person or Deity: Alternatively, the vertical line could depict a stylized human figure or a deity, with the upper branches signifying arms or headwear. The rounded element might then be interpreted as the body or a ceremonial object held by the figure.
Abstract Symbol: It's also possible that the character represents a more abstract concept, such as growth, stability, or connection between heaven and earth.
*Symbolic Possibilities:*
Fertility and Abundance: If associated with a tree or plant, the character might symbolize fertility, growth, and abundance.
Spiritual Connection: If representing a deity or human figure, it could signify communication with the divine, offerings, or prayers.
Stability and Groundedness: The vertical line and the rounded base could convey a sense of stability, rootedness, or a strong foundation.

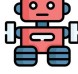 **GLM-4V**:
The character depicted is "甲" (jiǎ), which is one of the twenty-four ancient Chinese pictograms that form the basis for the modern Chinese script. It is pronounced like "jia" in English, with the first tone.
Literally, the character "甲" represents the upper part of a tortoise shell, from which it is derived. In ancient China, tortoise shells were used for divination, a practice known as "oracle bone divination." The cracks on the shell were interpreted as messages from the gods or spirits. Over time, these symbols became stylized into characters that could represent various concepts and words.
In its modern usage, "甲" has several meanings:
As an element in the periodic table, it stands for the chemical element potassium, with the atomic number 19.
In the context of the Chinese zodiac, "甲" represents the Year of the Tiger.
It can also be a prefix in Chinese words, indicating the first or foremost.
Additionally, "甲" is used in the Pinyin romanization system for Mandarin Chinese to represent certain tones and sounds.

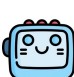 **Qwen-VL-Max**:
The character you have provided appears to be a combination of two or more elements, which is common in oracle bone script. The character seems to consist of a vertical line with a loop at the top and a horizontal line crossing it, with additional elements on either side. This structure is reminiscent of characters that might represent a specific object, action, or concept. Given the complexity and the limited understanding of oracle bone script, it is difficult to provide an exact translation without more context. However, characters with similar structures often relate to natural phenomena, agricultural practices, or religious rituals. For a more accurate interpretation, it would be helpful to know the context in which this character appears, such as the sentence or phrase it is part of, or the specific inscription it comes from. Additionally, consulting with a specialist in oracle bone script or using a comprehensive oracle bone script dictionary could provide more insight into the meaning of this character.

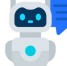 **MiniCPM-V 2.6**:
The oracle bone script in the image is a character that represents "plant" or "vegetation." In ancient Chinese divination practices, such characters were used to seek answers to questions posed during rituals. The specific design of this character suggests it could be related to agricultural themes or natural growth.

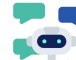 **Moondream2**:
The oracle bone character, which is depicted as a black silhouette in the image, is a symbol often associated with divination and fortune-telling. It is commonly used in various cultures and traditions to seek guidance and wisdom from the past or the future. The oracle bone symbolizes the connection between the present and the future, and it is often used in fortune tellers' services to provide insights and advice to those seeking guidance. The specific meaning of the oracle bone character may vary depending on the cultural context and the interpretation of the symbol.

**LLaVA-NeXT-8B**:
The image you've provided appears to be an oracle bone character, which is an ancient Chinese character carved on animal bones or turtle shells. Oracle bone characters were used for divination and were believed to have mystical powers. The specific meaning of this character would depend on the context in which it was used, as well as the interpretation of the diviner. However, without more information, it's difficult to provide a precise meaning. Oracle bone characters are now considered a part of Chinese history and are studied by scholars for their historical and linguistic significance.

Figure 16: A qualitative comparison for the deciphered results of 7 representative LMMs on a genuine undeciphered oracle bone character (III).

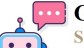 **OBI researcher**:
*Prompt (Prefix):*
You are a senior oracle bone researcher who excels in interpreting ancient texts. Please help me to predict the meaning of the undeciphered oracle bone characters reasonably and accurately.
*Prompt (Main):*
What is the meaning of this oracle bone character? *<image>* 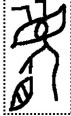

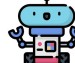 **GPT-4o**:
*Structural Elements:*
The top of the character features a structure resembling horns or a crown-like shape, often associated with deities, authority, or ritualistic symbolism. The central portion includes a circular or eye-like shape, which might represent vision, surveillance, or focus, often symbolizing observation or insight. The lower portion has a distinctive leaf or seed-like shape on one side and a curved line that could suggest movement or flow, possibly symbolizing growth, a sprout, or a natural element.
*Possible Interpretations:*
Symbol of Growth or Nature: The leaf or seed-like element indicates that the character might be connected to nature, growth, or fertility, possibly representing a plant, tree, or some organic process.
Divine or Ritual Context: The upper structure resembling horns or a crown could suggest a connection to a divine figure or ritual object, potentially indicating a sacred or ceremonial aspect.
Watchfulness or Insight: The eye-like shape could imply something related to seeing, monitoring, or knowing, hinting at a role of guidance, observation, or protection.
*Potential Meanings:*
The character may represent a concept tied to natural growth under divine guidance or protection, such as a sacred plant, a symbol of fertility, or a ritual object linked to agricultural or natural processes. It could also signify an entity or role connected to watching over growth, like a guardian spirit of plants or a symbolic overseer of fertility.

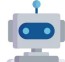 **Gemini 1.5 Pro**:
*Pictographic possibilities:*
Animal Representation: The overall shape with the prominent "eye" and protruding elements could suggest an animal, possibly a bird or an insect. The "striped" element at the bottom might signify claws, feathers, or a specific marking on the creature.
Human-like Figure: The central vertical line with extensions could also be interpreted as a stylized human figure, with the "eye" representing the head and the other elements indicating limbs or adornments.
*Symbolic interpretations:*
Spiritual Entity: The combination of animalistic and human-like features might point towards a supernatural being or a deity. The "eye" could symbolize watchfulness, knowledge, or divine power.
Natural Phenomenon: The character could also represent a celestial object like the sun or moon, or a weather phenomenon like lightning or rain. The "striped" element might indicate rays of light or falling rain. Abstract Concept: It could embody an abstract idea such as vision, perception, or transformation.

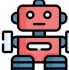 **GLM-4V**:
The character appears to be composed of two distinct parts:
The top part resembles a "hat" or headgear, which could imply a person of high status or authority, such as a king or noble. In Chinese culture, different headgear often signified rank or position.
Below the hat-like figure is a stylized representation of a bird in flight. Birds were commonly used in Chinese symbolism to represent freedom, transcendence, or spiritual journey.
Combining these elements, one might interpret this character as someone of high status who embarks on a significant spiritual or ceremonial journey.

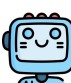 **Qwen-VL-Max**:
The character you provided appears to be a combination of several elements, which might represent a complex concept or a compound word. Here is a step-by-step analysis of the character:
1. Top Element: The top part of the character resembles a stylized representation of a plant or a tree. This could indicate a natural element or something related to agriculture or growth.
2. Middle Element: The middle part looks like a loop or a knot, which might symbolize something bound or connected. This could represent a concept like "to bind" or "to connect."
3. Bottom Element: The bottom part includes a leaf-like shape, which reinforces the idea of a plant or nature. The leaf might also suggest something related to life or vitality.
4. Overall Structure: The overall structure of the character, with its vertical and horizontal elements, might suggest a dynamic or active process, possibly related to growth, binding, or a natural cycle. Given these observations, a reasonable interpretation of this character could be related to concepts such as "binding growth," "natural connection," or "vital connection."

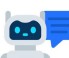 **MiniCPM-V 2.6**:
The oracle bone character depicted in the image is "若," which translates to "if" or "whether." This character has historical significance as it was used in ancient Chinese divination practices, particularly during the Shang dynasty. It represents a question posed in the context of seeking an answer from the gods through various rituals and sacrifices.

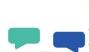 **Moondream2**:
The oracle bone character in the image is a symbol that represents the Chinese zodiac animal associated with the year of the person's birth. The symbol is a combination of a bone and a leaf, which is a common design in Chinese art. The meaning of the oracle bone character varies depending on the zodiac sign, but generally, it represents the year's characteristics and personality traits. For example, if the person born in the year of the oracle bone is a Leo, the symbol represents the zodiac sign of Leo and is often associated with confidence, leadership, and a strong sense of duty.

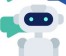 **LLaVA-NeXT-8B**:
The image you've provided appears to be an oracle bone character, which is a form of ancient Chinese writing. Oracle bone characters were used for divination and were carved onto animal bones, typically oxen, to predict the future. The specific meaning of this character would depend on the context in which it was used, as well as the interpretation of the diviner. Oracle bone characters are no longer used for divination in modern times but are still studied by scholars for their historical and linguistic significance.

Figure 17: A qualitative comparison for the deciphered results of 7 representative LMMs on a genuine undeciphered oracle bone character (IV).

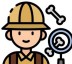 **OBI researcher**:
**Prompt (Prefix):**
You are a senior oracle bone researcher who excels in interpreting ancient texts. Please help me to predict the 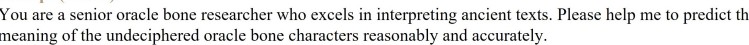
meaning of the undeciphered oracle bone characters reasonably and accurately.
**Prompt (Main):**
What is the meaning of this oracle bone character? *<image>*

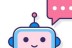 **GPT-4o**:
**Structural Elements:**
The left side of the character has a branched, tree-like structure with multiple protruding lines, resembling a plant, branch, or some form of organic growth. The right side appears to be a leaf or seed-like shape, which is common in Oracle Bone script to represent natural elements, seeds, or the concept of growth.
**Possible Interpretations:**
Nature and Growth: The tree-like structure on the left strongly suggests an association with plants, trees, or the broader concept of nature. This type of imagery is often linked to ideas of life, vitality, and regeneration.
Seed or Reproductive Element: The right side, resembling a leaf or seed, might symbolize fertility, potential, or the beginning of growth. In Oracle Bone script, such shapes are frequently used to denote something emerging or sprouting.
Symbolic Representation of a Cycle: The combination of tree-like and seed-like elements could indicate the cycle of life, from growth to reproduction, or a metaphorical connection to renewal and continuity.
**Potential Meanings:**
This character likely represents a concept related to growth, fertility, or the cyclical nature of life. It could symbolize a tree bearing seeds or the process of regeneration, possibly used in divinations concerning agriculture, fertility, or natural cycles. It may also be linked to a deity or spirit associated with plants or growth, used in rituals seeking favor for crops or natural abundance.

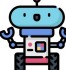 **Gemini 1.5 Pro**:
Given that the image doesn't resemble any known oracle bone script, I can only offer highly speculative interpretations based on its visual appearance:
**Pictographic Representation:** The left part could depict a person with raised arms, possibly engaged in an activity like dancing, celebrating, or praying. The right part, with its looped shapes, might symbolize an object or a natural phenomenon, such as a coiled rope, a swirling wind, or even a celestial body.
**Symbolic Interpretation:** The combination of the two elements might convey a concept related to human interaction with the environment or the spiritual realm. It could suggest a ritualistic dance, an offering to the gods, or a plea for favorable weather or a bountiful harvest.
Abstract Expression: The image might also represent abstract ideas or emotions. The dynamic posture on the left could symbolize joy, excitement, or aspiration, while the swirling forms on the right might evoke a sense of mystery, transformation, or the cyclical nature of life.

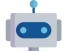 **GLM-4V**:
The character appears to be composed of two parts:
The top part looks like "火" (huo), which means fire in Chinese.
The bottom part seems to have been damaged or eroded, making it difficult to discern its original form. However, it might have been another character that has since become illegible due to time or damage.

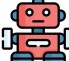 **Qwen-VL-Max**:
The character you have provided appears to be a combination of two distinct elements. Here is a reasonable interpretation based on the structure and known oracle bone characters:
1. The left part of the character resembles a hand or a gesture, which is often associated with actions or commands in oracle bone script.
2. The right part of the character looks like a stylized representation of a weapon or a tool, possibly a spear or an axe.
Combining these elements, the character might represent an action or command related to warfare or hunting. It could be a verb or a noun indicating something like "to fight," "to hunt," or "weapon." However, it's important to note that this is a speculative interpretation based on the visible elements and known patterns in oracle bone script. For a more accurate translation, consulting a comprehensive oracle bone script dictionary or an expert in the field would be necessary.

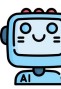 **MiniCPM-V 2.6**:
The oracle bone script is one of the earliest forms of writing in China. The character shown in the image, resembling a combination of strokes and loops, could be interpreted as a specific ideogram or symbol used in ancient Chinese divination practices. Oracle bones were inscribed with questions or comments by burning them on animal bones or turtle shells before being cracked open to receive answers from deities. Without additional context or knowledge of the specific sequence of characters, it's challenging to provide an exact meaning. However, such symbols often represented natural phenomena, celestial events, or concepts like life, death, and change. To accurately interpret this character, one would typically consult a scholar of ancient Chinese scripts or use a reference book that deciphers oracle bone inscriptions.

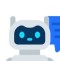 **Moondream2**:
The character depicted in the image is a form of the oracle bone script used in ancient China. This specific symbol, resembling a combination of lines and curves, could be interpreted as representing a concept or entity that combines elements of both the natural world and human creativity. Without additional context, it's challenging to determine its exact meaning, but it may relate to themes such as growth, change, or the blending of different forces or ideas. Oracle bones were often inscribed with questions or messages intended for divination, so this character might have been part of a larger set of symbols used in rituals or consultations.

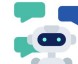 **LLaVA-NeXT-8B**:
The image you've provided appears to be a stylized representation of an oracle bone character. Oracle bone characters are ancient Chinese symbols that were used for divination and were carved onto animal bones, typically oxen, to predict the future. The specific character you've shown is not clearly legible due to the low resolution and stylization, but it seems to be a combination of two or more characters. To provide a more accurate interpretation, I would need a clearer image or more context. If you have a specific question about the character or its meaning, please provide more details or a clearer image, and I'll do my best to assist you.

Figure 18: A qualitative comparison for the deciphered results of 7 representative LMMs on a genuine undeciphered oracle bone character (V).

