# OpenReview forum: "OBI-Bench: Can LMMs Aid in Study of Ancient Script on Oracle Bones?"
_ICLR.cc/2025/Conference — ICLR 2025 Poster_

### Official Review · Reviewer_ZRWu · 2024-11-01

**Soundness:** 2
**Presentation:** 3
**Contribution:** 3
**Rating:** 5
**Confidence:** 3

**Summary:**

The paper introduces OBI-Bench, a benchmark designed to evaluate the capabilities of Large Multi-Modal Models (LMMs) in processing oracle bone inscriptions (OBI), an ancient form of Chinese writing. The benchmark includes 5,523 images sourced from diverse origins, covering five key tasks: recognition, rejoining, classification, retrieval, and deciphering. These tasks demand expert-level domain knowledge and cognitive abilities.

**Strengths:**

The paper presents OBI-Bench, a novel benchmark that systematically evaluates Large Multi-Modal Models on tasks related to oracle bone inscriptions, addressing an unexplored and complex domain in LMM research. This benchmark encompasses five critical tasks: recognition, rejoining, classification, retrieval, and deciphering, going beyond standard evaluations to include recovery and interpretation tasks；Designed to test LMMs with expert-level challenges, OBI-Bench requires deep domain knowledge and cognitive skills. This innovative approach pushes the boundaries of LMM capabilities, contrasting with typical benchmarks that focus on general performance； The benchmark is backed by a diverse dataset of over 5,500 images from archaeological findings, significant for training and evaluating LMMs in real-world applications. Additionally, the paper includes a comprehensive evaluation of both proprietary and open-source LMMs, providing valuable insights into their performance and limitations in deciphering ancient scripts, thus highlighting the potential for advancing research in historical linguistics and cultural heritage.

**Weaknesses:**

The paper introduces a dataset of 5,523 images, but its diversity and scale could be enhanced by incorporating more images from various sources and time periods. This expansion would improve the robustness of the benchmark and better reflect the complexities of OBI variations; The paper evaluates LMMs at a single point in time. A longitudinal study tracking the performance of LMMs over successive versions could provide valuable insights into the progress and potential of these models; The paper discusses the performance of LMMs on various tasks but does not delve into the interpretability of these models. Providing insights into how LMMs make decisions, especially on deciphering tasks, could be insightful for both the AI community and OBI scholars.

**Questions:**

Would the authors consider conducting a longitudinal study to track the performance of LMMs over time as the models evolve, to measure the progress and potential of these models in the OBI domain? How will the authors address the issue of cultural bias in LMMs, and will they investigate how different cultural backgrounds of the model's training data and developers might influence performance on OBI tasks?

---

> ### Author Response · Authors · 2024-11-15
> **Thank you for your valuable comments**
>
> We are grateful to the reviewer for acknowledging the significance of our findings and contributions! We hope the following clarifications can address your concerns.
>
> **The diversity and scale of the tested dataset.** As mentioned in Appendix D, we acknowledge that the current dataset is limited in size. For example, in the evaluation of deciphering task, we construct four evaluation scenarios including the perspective of character frequency, two character formations, and radical. In this paper, our goal is to use these representative scenarios to preliminarily explore the performance of LMMs, which provides a reference for more comprehensive evaluations in the future. This inevitably leads to the issue of sample size. It is important to note that, unlike the construction of other image datasets, collecting an oracle bone inscription dataset requires significant effort from domain experts (such as rejoining and deciphering). There are many Oracle datasets that are not actually publicly available, which prevents us from collecting richer OBI image content. In the future, we will continue to work with Oracle scholars to build large-scale open source Oracle datasets. Besides, we acknowledge that we will dynamically update and maintain this benchmark dataset to ensure its suitability for more robust future evaluations of LMMs.
>
> **A longitudinal study to track the performance of LMMs.**  We agree with your perspective that ensuring the timeliness of benchmark evaluations is crucial in the rapidly evolving era of LMMs. To address this, we selected 23 large models for testing in this study, spanning a year from the August 2023 to August 2024, to achieve as comprehensive an evaluation as possible. This selection aims to validate the relationship between the evolution of large models and their performance on oracle bone inscription tasks. Specifically, to track the performance of LMMs over successive versions, we evaluate the performance of LLaVA-v1.5-7B,  LLaVA-v1.5-13B, LLaVA-NeXT-8B, and LLaVA-NeXT-72B in terms of the release time and number of parameters. The results on five evaluated OBI tasks show a trend of model performance improving over time and with the increase in parameter size. In the future, we will carry on to conduct a longitudinal study to track the performance of LMMs, which will provide valuable insights for the future development of LMMs specifically designed for oracle bone inscriptions. To achieve this, we will open-source all tested data mentioned in this paper, which will be the first dataset specifically designed for validating the performance of LMMs in the field of Oracle Bone script research. Additionally, we will maintain and expand this dataset and benchmark dynamically.  For example, we plan to incorporate newly recognized rejoining or deciphering results from the Oracle Bone script field (ensuring compliance with copyright requirements). This holds significant value for the academic community (both the fields of humanities & history and computer science domains) and greatly contributes to expanding the application domains of LMMs.

---

> ### Author Response · Authors · 2024-11-15
> **Thank you for your valuable comments**
>
> **Further discussion on the interpretability of LMMs.** Thank you for your valuable suggestion. We agree that exploring the interpretability of LMMs is an important direction, especially for deciphering tasks related to OBI. In this paper, we adopt a natural-language explanations to elucidate model predictions [1, 2]. In Fig. 14, Fig. 15, Fig. 16, Fig. 17, and Fig. 18, we provide extra deciphering visualization of different LMMs. We can observe that GPT-4o and Gemini 1.5pro exhibit obvious logical reasoning process that first deconstructing the characters structurally, and then perform block-wise visual similarity analysis, followed by event or meaning prediction. Such strategy also explains why they achieve better decipherment performance in pictographic characters. While other LMMs especially those open-source LMMs exhibit relatively poor reasoning ability that either directly make a prediction about the character or lack the relevant knowledge background to provide an accurate response.
>
> Apart from these observations, here, we further provide some possible next steps of research related to the interpretability of LMMs. New possibilities of exploring the interpretability of LMMs in OBI tasks can be tried in a number of directions:
>
> - First of all, the simplest approach is using local explanations that providing feature attributions for input tokens, which reflects its impact on the model’s generated output.
> - Beyond this, chain-of-thought based prompting can explicitly reflects the reasoning ability of the LMMs in the process of deciphering.
> - Another way is to investigate the training data of the LMMs, which is relatively more difficult to reach. Essentially, how much oracle-related information is included in its training dataset determines its performance on OBI tasks.
> - Furthermore, it is also feasible to conduct a subjective study that explores the alignment of LMMs' reasoning processes with humans' perception. Comparing the inference steps of humans and LMMs in determining the meaning of OBIs can provide us with unique insights into more plausible forms of prompt, as well as the future development direction of LMMs dedicated to OBI tasks.
>
> In the final version, we will add these additional discussion in Appendix C.4.2.
>
> [1] Singh, Chandan, et al. "Rethinking interpretability in the era of large language models." *arXiv preprint arXiv:2402.01761* (2024).
>
> [2] Creswell, Antonia, Murray Shanahan, and Irina Higgins. "Selection-inference: Exploiting large language models for interpretable logical reasoning." *arXiv preprint arXiv:2205.09712* (2022).
>
> **The potential cultural impact of model's training data on OBI tasks.** We acknowledge that studying the cultural background of models and whether they exhibit biases when handling tasks with cultural context is one of the key directions of our research. However, as mentioned earlier, directly investigating the training datasets of LMMs is challenging, as few models publicly disclose their training sources. One possible approach is to explore the model's output preferences through task-oriented methods, which can help identify potential cultural biases that may exist.

---

> ### Author Response · Authors · 2024-11-23
>
> Dear Reviewer ZRWu,
>
> We sincerely appreciate the time and effort you've invested in reviewing our paper and providing valuable feedback. As we have now reached the midpoint of the author-reviewer discussion phase, we are eager to know if our rebuttal has effectively addressed your concerns. Your insights are extremely important to us.
>
> If you have any further questions or need additional clarification about our work, please feel free to reach out.
>
> Thank you once again for your time and consideration.
>
> Best regards

---

> > ### Author Response · Authors · 2024-12-02
> >
> > Dear Reviewer ZRWu,
> >
> > We truly appreciate your guidance to advance our work. We genuinely value the time and effort you dedicated to reviewing our paper. Considering that the discussion will end soon, we eagerly look forward to your response.
> >
> > Best regards,
> >
> > Authors

---

> > > ### Comment · Reviewer_ZRWu · 2024-12-03
> > >
> > > Thanks for the detailed rebuttal comment. I will maintain my current rating.

---

> > > > ### Author Response · Authors · 2024-12-03
> > > > **Thank the reviewer**
> > > >
> > > > We would like to thank the reviewer for all the valuable questions and suggestions. We are grateful for your engagement in the rebuttal process and keep the rating.

---

### Official Review · Reviewer_Qjaj · 2024-11-02

**Soundness:** 3
**Presentation:** 2
**Contribution:** 2
**Rating:** 6
**Confidence:** 4

**Summary:**

This paper focuses on an interesting research filed, Oracle Bone Script. Considering the understanding of Oracle Bone Script requires aligning its images with actual meanings, it indeed serves as a good evaluation of LLM's capabilities in vision-image alignment. The dataset consists of most of the currently available public datasets, covering five different tasks, namely recognition, rejointing, classification, retrieval, and deciphering. A detailed baseline is yielded among most proprietary and open-source LMMs.

**Strengths:**

1. The five domain problem is spanning from excavation to synthesis and coarse-grained to fine-grained which is novelty to the evaluation of a LMM.
2. The baseline consists of most proprietary and open-source LMMs and the results are consist to other LMM benchmarks.

**Weaknesses:**

1. Some domain problem setting, namely rejoining and deciphering, is not convincing.
2. Lack of the results of fine-tuned open-source LLMs which is quiet important to a domain specific benchmark.
3. Among the analysis of each domain problem or each scenario, the essential reason why open LMM perform like that is lacking which is important to the community beyond a specific benchmark.

**Questions:**

1. I am agree the rejoining problem is important to the recovery of paper money, calligraphy, and painting files. Does the rejoining problem is necessary in exploring the capability of understanding the OB of a LMM? I believe the rejoining problem only focus on the high-frequency image textures，but not the real meaning of a pattern. If so，a yes-or-not or how problem is rather superficial to evaluate this domain problem.

2. For the evaluating the performance of deciphering, a BERTScore might be bias. A detailed instruction prompt to identify the output patten of a LMM should be better.

3. It seems "yes-or-no" query and "how" query are alternative (according to the results and common sense). It is better to show these two queries are irrelevance.

4. Line 471. What does "with different frequency levels F" mean?

---

> ### Author Response · Authors · 2024-11-16
> **Thank you for your valuable comments**
>
> We thank the reviewer for the valuable comments and feedbacks. We are glad that the reviewer finds our work conducting comprehensive analysis, providing a good evaluation of LMM's capabilities in understanding oracle bone scripts. We provide details response as below.
>
> **Some domain problem setting is not convincing.** From the perspective of the history of OBI research, the timeline of rejoining can be divided into three phases based on significant milestones. The first phase spans from 1917, when Guowei Wang first successfully rejoined oracle bones [1], to 1938. The second phase begins in 1939, marked by Yigong Zeng's publication of a dedicated book on oracle bone rejoining [2], and continues until 1977. The third phase started in 1978 with the publication of *The Collection of Oracle Bone Inscriptions [3]*, which signaled a new era for oracle bone rejoining efforts.  —— according to Prof. Tianshu Huang in [4] (who is a leading authority and a renowned Chinese scholar specializing in ancient Chinese script studies)
>
> More recently, with the development of AI, numerous rejoining efforts based on computer technology have also emerged [5-9]. Here, we want to emphasize that since its discovery, the rejoining of OBI has consistently been an important research focus in the academic community. Whether through traditional manual rejoining methods or leveraging computer technology for automated rejoining, this task has remained at the forefront of scholarly efforts. To this end, we incorporated the sequential processing steps involved in the excavation and preservation of OBI and defined rejoining as a distinct research task for evaluating LMMs.
>
> Deciphering OBI has always been a significant area of research. It is essential for preserving cultural heritage, gaining historical insights, understanding language evolution, and fostering interdisciplinary research. Since its excavation and discovery, deciphering the meaning of OBI has been a goal pursued by many scholars in the fields of literature and history [10, 11]. In recent years, researchers in the field of computer science have also proposed numerous methods for deciphering OBIs [12-15].
>
> [1] Guowei Wang. "A Continuation of the Study on Ancestors and Kings Mentioned in Shang Oracle Inscriptions.", 1917
>
> [2] Yigong Zeng. "Jia Gu Zhuo Cun (The Oracle Bone Remaining Inscriptions Rejoined)", 1939.
>
> [3] Moruo Guo. Collections of oracle-bone inscriptions. The Zhong Hua Book Press, 1982.
>
> [4] Tianshu Huang. "The Academic Significance and Methods of Oracle Bone Rejoining." *Palace Museum Journal* 1 (2011): 7–13+156. doi:10.16319/j.cnki.0452-7402.2011.01.004.
>
> [5] Zhang, Weijie. "The importance of oracle rejoining in the study of ancient characters and the history of the Shang Dynasty." *Journal of Chinese Writing Systems* 3.1 (2019): 19-28.
>
> [6] Zhang, Chongsheng, et al. "Data-driven oracle bone rejoining: A dataset and practical self-supervised learning scheme." *Proceedings of the 28th ACM SIGKDD Conference on Knowledge Discovery and Data Mining*. 2022.
>
> [7] Tian, Yaolin, et al. "The research on rejoining of the oracle bone rubbings based on curve matching." *Transactions on Asian and Low-Resource Language Information Processing* 20.6 (2021): 1-17.
>
> [8] Zhang, Zhan, et al. "Deep rejoining model for oracle bone fragment image." *Asian Conference on Pattern Recognition*. Cham: Springer International Publishing, 2021.
>
> [9] Zhang, Chongsheng, et al. "AI-powered oracle bone inscriptions recognition and fragments rejoining." *Proceedings of the Twenty-Ninth International Conference on International Joint Conferences on Artificial Intelligence*. 2021.
>
> [10] Moruo Guo, "Studies in Ancient Chinese Society", 1930.
>
> [11] Xigui Qiu. Collected Works of Qiu Xigui. Fudan University Press, 2015.
>
> [12] Guan, Haisu, et al. "Deciphering Oracle Bone Language with Diffusion Models." ACL (2024).
>
> [13] Chang, Xiang, et al. "Sundial-gan: A cascade generative adversarial networks framework for deciphering oracle bone inscriptions." *Proceedings of the 30th ACM International Conference on Multimedia*. 2022.
>
> [14] Wang, Pengjie, et al. "An open dataset for oracle bone character recognition and decipherment." *Scientific Data* 11.1 (2024): 976.
>
> [15] Zhang, Gechuan, et al. "Deciphering Ancient Chinese Oracle Bone Inscriptions Using Case-Based Reasoning." *International Conference on Case-Based Reasoning*. Cham: Springer International Publishing, 2021.

---

> ### Author Response · Authors · 2024-11-16
> **Thank you for your valuable comments**
>
> **The essential reason why open LMMs perform poor?** To answer this question, we provide the following analysis according to the task category:
>
> - In the recognition task, the poor performance of open-source LMMs can be attributed to several reasons: 1. Open-source models generally have fewer parameters, which is crucial for obtaining precise and comprehensive descriptions of image content. 2. They perform poorly in locating specific oracle bone characters, and this issue appears to be independent of model size. This is because most of the evaluated LMMs lack functionality in this area. The InternVL2 series, in contrast, specifically incorporates grounding ability during training, resulting in relatively better performance.
> - In the rejoining, classification and retrieval tasks, the LMMs themselves do not provide specific prompts tailored for these tasks. The inferior of open-source LMMs mainly lie in their ability to produce precise or fine-grained judgments, which is directly related to the parameter size of language backbones.
> - In the OBI deciphering task, the performance of LMMs primarily depend on three key factors: the number of model parameters, the quality of its reasoning capabilities, and the comprehensiveness of its training dataset. Compared to proprietary LMMs, open-source LMMs are at a disadvantage in all these aspects. We can also observe from Fig. 14-18 that the step-by-step reasoning capabilities shown by GPT-4o and Gemini have significantly enhanced the accuracy and reliability of their deciphering results. This further highlights the necessity to enhance the reasoning ability of the LMMs in next-generation development.
>
> We will add these discussions to the final version of this paper as suggested.
>
> **The “yes-or-no” and “how” problem for the rejoining task.** We hold the opinion that the rejoining task for OBI goes far beyond simple image comparison. It is a sophisticated process that involves the integration of textual, physical, and historical knowledge:
>
> 1. OBI often contain fragmented texts distributed across multiple pieces. Rejoining requires understanding the linguistic and semantic patterns of the script to hypothesize potential matches.
> 2. In addition to textual content, the physical characteristics of the fragments, such as edge contours, thickness, and material texture are crucial.
> 3. Moreover, knowledge of the historical usage of Oracle bones, typical layouts of inscriptions, and stylistic variations can inform the rejoining process.
>
> Overall, we believe that traditional methods using computer vision or image matching for rejoining primarily focus on high-frequency image textures. In contrast, the reasoning process of LMMs involves a broader range of factors, including their visual perception capabilities, advanced cognitive abilities, and the influence of their knowledge base. Since there are currently no other similar LMM applications in this field, in the OBI-Bench, we evaluate the LMM’s performance via **“**yes-or-no” and “how**”** questions, which stand for a binary output and a probaility output. This reflects the visual-language capabilities of LMMs, as well as their ability to produce fine-grained evaluations or conclusions.
>
> **Bias for the deciphering task.** In the deciphering task, since there are no existing metrics for OBI deciphering, we use BERTScore to quantify the semantic similarity between the ground-truth and the output of LMMs.
> To adress the potential bias of BERTScore, we further calculate the cosine similarity metric. Here, we report the deciphering results on Tier-1 and pictograph sets for illustration. We list the evaluated rank of LMMs to see whether BERTScore introduces evaluation bias (Rank of BERTScore/Rank of cosine similarity):
> | LMM| Tier-1| Pictograph|
> |-------|-------|-------|
> |GEMINI 1.5 PRO|2/2|3/3|
> |GEMINI 1.5 FLASH|4/4|4/4|
> |GPT-4V|3/3|2/2|
> |GPT-4O|1/1|1/1|
> |QWEN-VL-MAX|5/5|5/5|
> |GLM-4V|6/6|6/6|
> |-------|-------|-------|
> |xGen-MM|11/10|8/8|
> |mPLUG-Owl3|6/7|3/3|
> |MiniCPM-V 2.6|8/8|6/6|
> |Moondream2|13/16|10/11|
> |InternVL2-Llama3-76B|2/2|2/2|
> |InternVL2-40B|3/3|4/4|
> |InternVL2-8B|4/4|5/5|
> |GLM-4V-9B|10/11|15|
> |CogVLM2-Llama3-19B|14/14|13/14|
> |LLaVA-NeXT-72B|7/6|9/9|
> |LLaVA-NeXT-8B|15/15|16/16|
> |IDEFICS-2-8B|5/5|7/7|
> |DeepSeek-VL|17/17|17/17|
> |InternLM-XComposer2-VL|1/1|1/1|
> |LLaVA-v1.5-13B|9/9|11/10|
> |LLaVA-v1.5-7B|16/13|14/12|
> |Qwen-VL|12/12|12/13|
>
> As shown in this Table, we notice that the results of BERTScore and cosine similarity are the same in evaluating proprietary LMMs, since their performance is significantly different. However, in cases where the performance difference is minimal, such as comparing LLaVA-v1.5-7B and Moondream2, the slight variation in LLaVA's output words leads to a small change in cosine similarity, failing to accurately reflect actual performance differences. In the future, we will improve this evaluation strategy by adopting a detailed instruction prompt to identify the output pattern of a LMM.

---

> ### Author Response · Authors · 2024-11-16
> **Thank you for your valuable comments**
>
> **Lack of the results of fine-tuned LMMs.**  First, fine-tuning a LMM requires a substantial corpus specifically related to OBI tasks. Referring to other specialized fields, such as medicine [1] and law [2], the amount of data used for fine-tuning large language models typically exceeds 100k instances. Second, the ancient character images and linguistic interpretations related to Oracle Bone script are much scarcer and more specialized compared to those fields. Additionally, there is currently no existing database in this area, which presents significant challenges for further exploring the capability of fine-tuning large models in this domain. Currently, the absence of a comprehensive corpus in OBI domain severely limits the applicability of language models. The existence of these challenges also motivates us to work towards building a large-scale, dedicated OBI corpus in the future.
>
> [1] Li, Yunxiang, et al. "Chatdoctor: A medical chat model fine-tuned on a large language model meta-ai (llama) using medical domain knowledge." *Cureus* 15.6 (2023).
>
> [2] Fei, Zhiwei, et al. "Lawbench: Benchmarking legal knowledge of large language models." *arXiv preprint arXiv:2309.16289* (2023).
>
> **Are "yes-or-no" query and "how" query alternative?** First, as introduced in Sec. 2.2.2, we design four question types, What, Yes-or-No, How, and Where, to simulate multiple human queries in various OBI tasks, each tailored to different levels of granularity from both the question and answer perspectives. Their purposes and meanings are inherently different. In the recognition task, the Yes-or-No question is used to obtain the overall judgement of LMMs to the evaluated image, while the How question is used to obtain more fine-grained visual perception results, namely, the number of oracle bone characters in the evaluated images. In the rejoining, classification, and retrieval tasks, we use Yes-or-No and How questions to generate binary and probability output due to the characteristics of tasks. This reflect the difference performance of LMMs when being prompted by different types of query, providing insights (such as the design of CoT) in improving the performance for handling OBI tasks. Similar settings have been used in evaluating the low-level visual perception ability of LMMs [1, 2], LLM’s behavior [3] or other more general evaluations [4].
>
> [1] Wu, Haoning, et al. "Q-Bench: A Benchmark for General-Purpose Foundation Models on Low-level Vision." *The Twelfth International Conference on Learning Representations (2024)*.
>
> [2] Wu, Haoning, et al. "Towards open-ended visual quality comparison." *European Conference on Computer Vision*. Springer, Cham, 2025.
>
> [3] Abbasiantaeb, Zahra, et al. "Let the llms talk: Simulating human-to-human conversational qa via zero-shot llm-to-llm interactions." *Proceedings of the 17th ACM International Conference on Web Search and Data Mining*. 2024.
>
> [4] Zhuang, Honglei, et al. "Beyond yes and no: Improving zero-shot llm rankers via scoring fine-grained relevance labels." NAACL 2024.
>
>
> **The meaning of "with different frequency levels F" in Line 471.**  The ‘F' in Line471 denotes the character frequency of the Oracle bone inscript, which is categorized according to two authoritative books on Oracle Bone script research [1, 2]. For example, Tier-1: F≥500 means that the character appears more than 500 times among all unearthed Oracle Bone inscriptions, so it was classified as a ‘Category I commonly used character’.  We will refine the corresponding description in the final version to enhance the readability of the paper.
>
> [1] Zhiji Liu. On the two concentration features of the character frequency of bone inscriptions. Studies in Language and Linguistics, 30(04):114–122, 2010.
>
> [2] Tingzhu Chen. Restudy of the structural system of the Yin Shang oracle bone inscriptions. Shanghai People’s Publishing House, 2010.

---

> > ### Comment · Reviewer_Qjaj · 2024-11-20
> > **upgrade rate**
> >
> > Thanks for the detailed rebuttal comment. Author's feedback addressed most of my concerns and I am glad to change my rate to 6.

---

> > > ### Author Response · Authors · 2024-11-20
> > > **Thank the reviewer**
> > >
> > > We would like to thank the reviewer for all the valuable comments and questions. We are grateful for your engagement in the rebuttal process and the final upgrade of the rating.

---

### Official Review · Reviewer_x4po · 2024-11-02

**Soundness:** 3
**Presentation:** 3
**Contribution:** 3
**Rating:** 6
**Confidence:** 3

**Summary:**

This paper introduces OBI-Bench, a benchmark designed for evaluating multiple capabilities in the field of oracle bone script research. OBI-Bench encompasses five domains: recognition, rejoining, classification, retrieval, and deciphering. It involves four types of questions and includes a total of 5,523 questions, ultimately demonstrating these capabilities through extensive experiments.

**Strengths:**

1. This paper covers five specific areas of oracle bone script exploration, providing a comprehensive summary of prior work in the field.
2. The evaluation dimensions of this paper are diverse, it attempts to explore more fine-grained capabilities through the design of questions such as "How" and "Where" questions. Although the design of these two types of questions may not be critical, this approach seems to offer a potential framework for exploring process supervision mechanisms for models in the field of oracle bone script research.

**Weaknesses:**

1. There are some long-tail issues in the data volume of each task, particularly with an excessive amount for Recognition and insufficient data for Deciphering.
2. Despite the diversity of tasks, would it be possible to provide an overall score to evaluate the comprehensive ability of LMMs? This score should not be a simple average but should also take into account that LMMs are still in the early stages in the field of oracle bone scripts. Consideration should be given to the lower scores that may result from the model not being trained in certain areas. It would be beneficial to design a dynamic evaluation score to provide a more comprehensive guide for LMMs.

**Questions:**

Please refer to the weaknesses section.

---

> ### Author Response · Authors · 2024-11-15
> **Thank you for your valuable comments**
>
> We thank the reviewer for the positive rating of our paper! We also appreciate the reviewer for acknowledging the novelty of this work and all the constructive suggestions. We hope the following clarifications can address the reviewer's concerns.
>
> **Long-tail issues in the data volume.**  As mentioned in Appendix D, we acknowledge that the current dataset is limited in size and exists long-tail issues. On the one hand, due to the different nature of the task, the number of datasets for recognition is much larger than rejoining and deciphering tasks, which is directly determined by the amount of domain expertise required for the task and the workload of domain experts. On the other hand, from the content contained in the dataset itself, nearly all representative OBI datasets mention the long-tail problem [1, 2, 3], i.e., the distribution of instances against their categories. Due to the scarcity of resources and the problem of archaeological excavations, many characters only contain a small number of instances. In order to achieve a reasonable evaluation task design, we have to optimize the used data structure. To further address this problem, we plan to open-source all tested data mentioned in this paper, which will be the first dataset specifically designed for validating the performance of LMMs in the field of OBI research. In addition to the test sets collated from existing datasets, we also constructed the Original Oracle Bone Recognition (O2BR) dataset (Appendix.B.1) and the OBI-rejoin dataset (Appendix.B.2) to fill gaps in this field. Additionally, we will maintain and expand these datasets and benchmark dynamically to facilitate the academic community to improve this issue, and hopefully to collaborate with other peers to achieve material sharing.
>
> [1] Huang, Shuangping, et al. "OBC306: A large-scale oracle bone character recognition dataset." *2019 International Conference on Document Analysis and Recognition (ICDAR)*. IEEE, 2019.
>
> [2] Guo, Jun, et al. "Building hierarchical representations for oracle character and sketch recognition." *IEEE Transactions on Image Processing* 25.1 (2015): 104-118.
>
> [3] Wang, Pengjie, et al. "An open dataset for oracle bone script recognition and decipherment." *arXiv preprint arXiv:2401.15365* (2024).
>
> **Would it be possible to design a dynamic evaluation score?** Considering the diversity of tasks and the capabilities of LMMs involved (both visual perception and high-level cognitive ability), currently, we are more inclined to adopt an objective evaluation algorithm for LMMs based on task performance. We also agree with the reviewer that LMMs that have not been trained in particular domains are more inclined to reap lower scores. Here, we also present a scheme for designing such objective evaluation algorithms in the future:
>
> - **Task-Weighted Scoring**: Given the varying importance and difficulty of different tasks, different weights can be assigned to each task, which should be designed based on the needs of OBI researchers.
> - **Hierarchical Scoring Framework**: Establish different tiers of abilities, such as basic recognition, semantic understanding, and background knowledge application. Scores can be calculated individually for each tier and then combined to yield a more comprehensive overall score. This allows for more flexibility in capturing the model's performance across diverse tasks.
> - Considering that LMMs are still in the early stages in the field of processing OBI, **a dynamic adjustment mechanism** can be introduced.  A “maturity” factor can be incorporated to reflect the model’s gradual learning and adaptation within this domain. This dynamic mechanism makes the score more forward-looking, capturing the model’s potential for improvement in areas where it is less developed. However, given the dearth of oracle corpora, it is difficult to compare its performance after and before fine-tuning. This also provides motivation for us to further build large-scale task-oriented oracle corpora or question-answer pairs.

---

> ### Author Response · Authors · 2024-11-23
>
> Dear Reviewer x4po,
>
> We sincerely appreciate the time and effort you've invested in reviewing our paper and providing valuable feedback. As we have now reached the midpoint of the author-reviewer discussion phase, we are eager to know if our rebuttal has effectively addressed your concerns. Your insights are extremely important to us.
>
> If you have any further questions or need additional clarification about our work, please feel free to reach out.
>
> Thank you once again for your time and consideration.
>
> Best regards

---

> > ### Author Response · Authors · 2024-12-02
> >
> > Dear Reviewer x4po,
> >
> > We truly appreciate your guidance to advance our work. We genuinely value the time and effort you dedicated to reviewing our paper. Considering that the discussion will end soon, we eagerly look forward to your response.
> >
> > Best regards,
> >
> > Authors

---

> > > ### Comment · Reviewer_x4po · 2024-12-03
> > >
> > > Thank you for your response. It addresses part of my concerns.
> > >
> > > However, I believe a comprehensive oracle bone benchmark should aim to design tasks for each character, enabling it to address the long-tail distribution problem and provide a more complete evaluation of model performance. Admittedly, this is a challenging task, with no prior work to reference.
> > >
> > > Therefore, I will maintain my current rating.

---

> > > > ### Author Response · Authors · 2024-12-03
> > > > **Thank the reviewer**
> > > >
> > > > We would like to thank the reviewer for all the valuable questions and suggestions. We are grateful for your engagement in the rebuttal process and keep the positive rating. We will take this framework as a longitudinal study to track the performance of LMMs and continuously improve the evaluation.

---

### Official Review · Reviewer_7Yec · 2024-11-11

**Soundness:** 3
**Presentation:** 3
**Contribution:** 3
**Rating:** 8
**Confidence:** 3

**Summary:**

This paper introduces OBI-Bench, a comprehensive benchmark for evaluating large multi-modal models (LMMs) on oracle bone inscription (OBI) processing tasks. The work aims to assess whether modern AI models can assist in studying and interpreting these important historical artifacts, by repositioning the LLMs as subject experts for the tasks.  The evaluation targets critical domain challenges in: recognition, rejoining, classification, retrieval, and deciphering.

The paper highlights the use-cases of the current LMMs, and also throws caution towards the issues.

**Strengths:**

1. Novel (and important) Application Domain - Important systematic evaluation of LMMs for ancient script analysis, addressing a significant real-world problem in historical research with potential to accelerate archaeological research and cultural heritage preservation.

2. Comprehensive fine-grained benchmark - covering tasks like recognition, rejoining, classification, retrieval, and deciphering

3. Specific data curation for each task helps to answer queries specific to each of them.

4. Extensive Evaluation with 23 state-of-the-art LMMs (6 proprietary, 17 open-source)

**Weaknesses:**

1. The overall technical importance of the benchmark might be limited - the evaluation is very sensitive to the query-answer form (spectrum of questions), even a little change or adding some context can influence the outcomes drastically. Although the work is quite extensive, a decent foray into how prompt engineering can impact the performance could have been a nice addition to the work. Eg: Take the example of Deciphering task, use the best open-source and proprietary models (as already done in evaluation for this task) and test this for prompt engineering.

2. How many images are original and how many are pseudo-oracle bone characters?  What are the differences in evaluation between both these types of bone-fragments? - these are unclear and not mentioned in the main article.

3. Section 3.1 (scenario 2): What about the situation when the query does not include the word ‘OBI‘ (reflection of first point above)?

4. An interesting trend is visible in Fig. 3, can there be another experiment to check if this trend is correlated with the parameter sizes? Eg: Fix a type of LMM, and then observe the effect of increasing parameters?

5. Tab. 5 - Recall@10 is roughly 10x Recall@1 which is understandable. More meaningful will be to have Recall@3 and/or Recall@5 - often (like in Google search results) users aren’t interested in top-10 recalls, rather how good is the top-3/top-5.

6. Some discussion on the impact of model architectures (of course only possible with open-source models), and computational requirements can improve the overall comprehensiveness of the work.

**Questions:**

1. What’s the main takeaway of the OBI-Benchmark? Having presented the empirical studies in the paper, it is clear that LMMs are sensitive to local/input information (borders, fragments, low-resolution, noisy fragments), perform better when the data is clean with better resolution, and are short on domain specific knowledge. In most of these tasks, better data will improve the performance but that is the nature of the OBI data…it is quite difficult to get a higher quality. So, as a researcher what shall I do? -- A discussion or propositions in this direction could be useful to add in the paper.

2. Non-generative AI models have recently become more interpretable and explainable [1]. So, given the niche domain of the data handled in OBI-Bench, wouldn’t it be more useful if such traditional ‘deep learning‘ methods are used if there is better data available, as compared to using LMMs which (as pointed above and in the article) are not at par and also are still a bigger black-box model?

3. Stress-testing the LMMs by increasing no. of categories clearly shows the real limitations of the LMMs. Given this finding (Fig. 3), do the authors think scaling is an issue? - that the performance will improve if the models were bigger with larger context windows?

4. There is a “Human“ performance metric mentioned in Table.2, is it possible to have such reference metrics for consequent tables as well? This should help understand the performance gap of the LMMs vs. Humans.

[1] Minh, D., Wang, H.X., Li, Y.F. et al. Explainable artificial intelligence: a comprehensive review. Artif Intell Rev 55, 3503–3568 (2022). https://doi.org/10.1007/s10462-021-10088-y

---

> ### Author Response · Authors · 2024-11-17
> **Thank you for your valuable comments**
>
> We thank the reviewer for the positive rating of our paper! We also appreciate the reviewer for acknowledging the novelty of this work and all the constructive suggestions. We hope the following clarifications can address the reviewer's concerns.
>
> **The technical importance of the benchmark.** We sincerely appreciate your insightful feedback and suggestions. We acknowledge that the evaluation might be sensitive to the query-answer form, as even small changes or added context may influence the outcomes. This is an important point, and we agree that it highlights the complexity and challenges inherent in evaluating LMMs. To this end, we have investigated the effect of pre-prompt in Appendix.C.4.1 where we provide specific instructions, contextual information, or role settings that allow the model to understand the user’s needs and expectations and thus provide a more accurate and relevant response. Here, we add more experiments to explore the impact of prompt engineering in other four OBI tasks. As suggested, we select the average best proprietary (GPT-4o) and open-source (InternVL2-Llama3-76B) LMMs for illustration.
>
> We follow the instructions in [1, 2] that there are some steps involved in creating effective prompts: 1.  Defining the goal; 2. Understanding the model’s capabilities; 3. Choosing the right prompt format; 4. Providing context.
>
> We thereby add a task-oriented role assignment prompt before our designed queries for different tasks.
>
> \# System: “You are a senior oracle bone researcher who excels in <{recognizing, rejoining, classifying, retrieving}> ancient texts. Please help me to <action>:
>
> - <action1>: recognize the content in this image reasonably and accurately.
> - <action2>: determine whether these oracle bone fragments can be successfully rejoined together reasonably and accurately.
> - <action3>: classify these oracle bone characters reasonably and accurately.
> - <action4>: classify these oracle bone characters reasonably and accurately.
>
> Note that the action keywords in classification and retrieval are the same due to the intrinsically correlated nature of the two tasks. Moreover, we further introduce the task-oriented case instuction:
>
> - recognition: \# System: This is a original oracle bone inscription with <number> characters on it. Their positions in the image are <[x1,y1,x2,y2], …., [xn,yn,xn+1,yn+1]>
> - rejoining: \# System: Here are three pairs of rejoined oracle bone fragments. <image1> and <image2> are connected, ………….
> - classification and retrieval: # System: Here are three pairs of oracle bone characters that belong to different categories. <image1> and <image2> belong to the same category, and their meaning is <meaning>, ………….
>
> **The results for the recognition task:**
> | LMM| O2BR-What| O2BR-Yes-or-No|O2BR-How|O2BR-Where| YQWY-What| YQWY-Yes-or-No|YQWY-How|YQWY-Where|
> |-------|-------|-------|-------|-------|-------|-------|-------|-------|
> |GPT-4o|0.6114|99.95%|0.4016|0.0038|0.3734|99.90%|0.3458|0.0182|
> |GPT-4o+Prompt Engineering|0.7986|99.95%|0.3975|0.0042|0.4724|99.95%|0.3483|0.0194|
> |-------|-------|-------|-------|-------|-------|-------|-------|-------|
> |InternVL2-Llama3-76B|0.5833|99.65%|0.4328|0.0976|0.3892|99.75%|0.5344|0.0623|
> |InternVL2-Llama3-76B+Prompt Engineering|0.6025|99.75%|0.4276|0.0978|0.4749|99.75%|0.5234|0.0635|
>
> **The results for the rejoining task:**
> | LMM|Yes-or-no| Acc@1|Acc@5|Acc@10|
> |-------|-------|-------|-------|-------|
> |GPT-4o|32.21%|29.13%|48.43%|78.47%|
> |GPT-4o+Prompt Engineering|34.26%|29.80%|48.46%|79.62%|
> |-------|-------|-------|-------|-------|
> |InternVL2-Llama3-76B|20.13%|5.18%|16.98%|31.68%|
> |InternVL2-Llama3-76B+Prompt Engineering|22.78%|13.28%|19.36%|37.84%|
>
> **The results for the classification task:**
> | LMM|Yes-or-no| Acc@1|Acc@5|
> |-------|-------|-------|-------|
> |GPT-4o|72.75%/74.50%/62.50%|89.75%/90.25%/75.50%|100.0%/100.0%/93.50%|
> |GPT-4o+Prompt Engineering|73.75%/74.75%/63.50%|89.75%/90.25%/75.50%|100.0%/100.0%/93.50%|
> |-------|-------|-------|-------|
> |InternVL2-Llama3-76B|44.75%/47.50%/43.25%|53.75%/55.00%/50.75%|69.75%/69.00%/66.75%|
> |InternVL2-Llama3-76B+Prompt Engineering|46.75%/48.50%/46.75%|53.75%/55.05%/50.75%|70.05%/69.00%/66.75%|

---

> ### Author Response · Authors · 2024-11-17
> **Thank you for your valuable comments**
>
> **The results for the retrieval task:**
> | LMM|Yes-or-no| Recall@1|Recall@3|Recall@10|mAP@5|
> |-------|-------|-------|-------|-------|-------|
> |GPT-4o|0.4550/0.6122|0.235/0.250|0.686/0.706|1.000/1.000|0.688/0.80|
> |GPT-4o+Prompt Engineering|0.4696/0.6226|0.235/0.250|0.686/0.706|1.000/1.000|0.688/0.80|
> |-------|-------|-------|-------|-------|-------|
> |InternVL2-Llama3-76B|0.3557/0.4268|0.150/0.250|0.460/0.675|1.000/1.000|0.522/0.72|
> |InternVL2-Llama3-76B+Prompt Engineering|0.3628/0.4339|0.150/0.250|0.460/0.675|1.000/1.000|0.528/0.74|
>
> We gain three valuable observations from these Tables. **First**, applying prompt engineering significantly increase the accuracy of answers under coarse-grained ‘what’ query in recognition task, whereas it is nearly ineffective in fine-grained "how" and "where" tasks that require detailed visual capabilities. **Second**, applying prompt engineering is more effective under ‘Yes-or-No’ query than ‘how’ question in rejoining, classification, and retrieval tasks. We suspect this is because LLMs themselves struggle with generating fine-grained comparative outputs (similar to the conclusion in the main paper). **Third**, applying prompt engineering to optimize LMM’s performance in specific tasks can only marginally improve the accuracy of model output patterns, without effectively enhancing perceptual abilities in visual. Its effectiveness is more significantly limited by the lack of domain-specific knowledge.
>
> [1] Marvin, Ggaliwango, et al. "Prompt engineering in large language models." *International conference on data intelligence and cognitive informatics*. Singapore: Springer Nature Singapore, 2023.
>
> [2] White, Jules, et al. "A prompt pattern catalog to enhance prompt engineering with chatgpt." *arXiv preprint arXiv:2302.11382* (2023). (Over 1100 citations)
>
> **Details for different types of OBIs.  I**n this paper, we categorize the appearance of OBI into five types, including original oracle bone, inked rubbing, oracle bone fragments, cropped sample, and handprinted OBI (Fig. 1). Among them, original oracle bone, inked rubbing, oracle bone fragments, and cropped sample belong to original oracle bone characters while handprinted OBIs (namely the pseudo-oracle bone characters) refer to replicas or imitations of OBIs that are created by hand rather than being authentic ancient inscriptions. These reproductions are often made by scholars, artists, or enthusiasts to study the script's structure, strokes, and overall appearance. As shown in Tab. 1, the datasets HWOBC and Oracle-50K used in the classification task as well as all datasets used in deciphering task are handprinted (1,140 pseudo-oracle bone characters). All other datasets are original (4,383). Original oracle bone were normally processed through the manual rubbing and image cropping processes for preservation. After thousands of years of natural weathering, corrosion, and man-made destruction, there are various types of noise in oracle bone rubbings, which may increase the difficulty of model perception and recognition.
>
> The principle for taxonomy lies in aligning the process of OBI from excavation to preservation with its corresponding phased tasks. For example, it is unnecessary to perform localization and recognition on handprinted OBIs. We will add more descriptions about the differences in evaluation between both these types of OBI in **Sec.2.2.1-Axis2** as suggested.
>
> **Question about Section 3.1 (scenario 2).** In scenario 2, we use the “Yes-or-No” query after specifying the response direction (the words “OBI” appear explicitly in the query), namely “Is this an oracle bone inscription image”. This is to supplement the scenario 1 that the words oracle bone do not appear explicitly in the “what” query, namely “What is in this image?” In this way, we can compare the visual perception ability of LMMs before and after a given target keyword.
>
> **The results of increasing the number of categories & More experiments for Fig. 3.** As suggested, we conduct extra experiments to investigate the performance trends correlated with the parameter sizes. Specifically, InternVL2-8B, InternVL2-40B, InternVL2-76B, GPT-4v are selected for comparison.  Please see the revised Fig. 3 and the corresponding discussions in the uploaded revised paper. The results show that increasing model size can to some extent improve the performance of the model when facing comparative tasks. This conclusion also applies in the field of convolutional neural networks.
>
> **Recall@5.** We add the results of Recall@3 for Tab. 5 as suggested. Please see the uploaded revised paper (**blue font** denotes the added contents/results)

---

> ### Author Response · Authors · 2024-11-17
> **Thank you for your valuable comments**
>
> **The impact of model architectures.**  We list the detailed information for 17 open-source LMMs in Tab.9. We notice that the evaluated LMMs own a decoder-only language architecture while differs in the parameter sizes. In the realm of vision backbones and vision-language alignment, several LMMs employ similar structures. Therefore, the main computational problem in OBI-Bench is the deployment of LMMs, since it requires GPUs with large memory (such as A100-80G, A800-80G, or multiple RTX4090). In this paper, we have not evaluated the internal structure or modules of LMMs for their impact on the performance of OBI processing, as it is not the objective of this study. Instead, we treat them as black boxes and conduct input-output evaluation. Thank you for your thoughtful and perceptive feedbacks. We acknowledge that this will be a significant focus of our future research efforts.
>
> **What’s the main takeaway of the OBI-Bench to the researcher?** The main takeaway of the OBI-Benchmark is to establish a comprehensive framework for evaluating the performance of both human and LMMs on tasks related to OBI, revealing both their potential and limitations. As the reviewer pointed out, the inherent challenges of OBI data, such as fragmentation, noise, and low resolution, are fundamental to this domain and cannot be fully avoided. However, these challenges also present opportunities for researchers to innovate. We plan to include a discussion section in the paper with the propositions for researchers:
>
> - **Focus on Robust Preprocessing Pipelines:** Developing preprocessing techniques that can enhance low-quality or noisy OBI data, such as denoising, fragment reconstruction, or super-resolution methods, may significantly improve LMM's performance on these tasks.
> - **Domain-Specific Fine-Tuning with Interdisciplinary Collaboration:**  Fine-tuning models on domain-specific datasets can help bridge the gap in domain knowledge. Integrating visual and textual modalities can enrich the understanding of oracle bone inscription, thus enhancing the overall interpretative capability. Bridging the gap between computer science and humanities disciplines is crucial for effective application of LMMs in OBI studies. Collaborating with historians and linguists to create multi-modal annotated OBI datasets may benefit this effort.
> - **Interactive Question Answering Systems:** Future applications could include interactive question-answering systems where users can inquire about oracle bone inscription meanings and historical contexts in natural language, leveraging large multi-modal models for intuitive and accessible interactions.
>
> It is hoped that these can provide actionable ideas for researchers and would trigger further discussion, and more importantly, new exploration in this field. Please see the newly added Appendix.E in the uploaded revised paper.
>
> **Why benchmarking LMMs rather than using Non-generative AI models?** In the current field of OBIstudies, research predominantly revolves around deep learning models, which vary in type according to task requirements, with no single non-generative AI capable of unifying all tasks. With the ongoing development of LMMs, their multimodal capabilities in visual-language are rapidly advancing. We envision a broader perspective for the future, whereby natural language question answering could potentially address applications in OBI and other ancient text processing. For instance, using LMM as the core processing unit of textual systems offers a more interactive experience compared to non-interactive deep learning or retrieval algorithms. People can obtain answers through human-like communication patterns, thereby reducing the learning curve for traditional humanities scholars and the general public who lack a strong computer science background required for model training. We hope our work can serve as a reference point for discussions on developing domain-specific multi-modal foundation models towards ancient language research and enhancing the untapped potentials of LMMs.
>
> **The performance of “Human”.** In this paper, we report human performance on OBI recognition and deciphering tasks for comparison, as we consider these two tasks to be primarily human-driven. In contrast, other tasks such as classification and retrieval, which are more commonly used in the construction of large-scale OBI databases, are predominantly model-driven. Therefore, we did not include statistics on human performance for these tasks. Note that collecting a large number of subjective results requires significant time and effort. At this stage, it is challenging for us to gather additional large-scale human evaluations. Thank you for your suggestion. In the future, we will further explore the performance differences between expert-level humans, the general public, and LMMs in greater detail.

---

### Meta-Review · Area_Chair_8Jck · 2024-12-21

**Metareview:**

a) This paper introduces a benchmark for evaluating large multi-modal models (LMMs) on oracle bone inscription processing tasks. It includes four types of questions on five domains: recognition, rejoining, classification, retrieval, and deciphering. A detailed evaluation with most proprietary and open-source LMMs is performed.

b) Ancient script analysis with LLM is a novel domain of application with potential to accelerate archaeological research and cultural heritage preservation. A comprehensive fine-grained benchmark with over 5500 annotated images and five different tasks. Extensive Evaluation with 23 different LMMs.

c) Revs. asked for clarifications and some additional experiments. For instance, rev. x4po complained about the imbalance between recognition and deciphering tasks and asked for a better evaluation protocol for the LLMs. Rev. ZRWu asked to incorporate more images from various sources and time periods and perform a longitudinal study tracking the performance of LMMs over successive versions.

d) Overall I think that the paper deserves publication because all reviewers agreed about the quality of the contribution, and the found problems are minor or corrected in the rebuttal.

**Additional Comments On Reviewer Discussion:**

Rev. 7Yec is the most positive, but she/he asked also many important questions, such as the sensitivity to the questions form or whether the proposed images were new or derived form other sources. Rev. was satisfied form the authors rebuttal and increased their score to 8.

Rev. x4po asked about long tail issues (much more recognition tasks than deciphering) and proposed a different evaluation of the LLMs performance. The answers by the authors did not fully satisfied rev, but she/he admitted that the task is new and there is no reference to get inspiration from. She/he maintained their score of 6.

Rev. Qjaj asked many pertinent questions about possible biases and problems in the evaluation. Responses form the authors were satisfactory and rev. increased their score to 6.

 Rev. ZRWu asked to incorporate more images from various sources and time periods and perform a longitudinal study tracking the performance of LMMs over successive versions. While this can be interesting, I do not think it is necessary for the paper. After the rebuttal, rev. said that she/he will maintain their score of 5, without explaining the reasons.

Most reviewers are happy about the paper and believe that the introduction of the proposed new task, dataset and benchmark are important contributions. Rev. ZRWu scored the paper with 5, however, their comments were more about propositions to improve the paper rather than critiques. Thus, overall I think the paper deserves publication.

---

### Decision · Program_Chairs · 2025-01-22

Accept (Poster)